# A conserved tooth resorption mechanism in modern and fossil snakes

A. R. H. LeBlanc [1,2] ✉, A. Palci [3,4], N. Anthwal [5], A. S. Tucker [5], R. Araújo [6], M. F. C. Pereira[7] & M. W. Caldwell[1]

Whether snakes evolved their elongated, limbless bodies or their specialized skulls and teeth first is a central question in squamate evolution. Identifying features shared between extant and fossil snakes is therefore key to unraveling the early evolution of this iconic reptile group. One promising candidate is their unusual mode of tooth replacement, whereby teeth are replaced without signs of external tooth resorption. We reveal through histological analysis that the lack of resorption pits in snakes is due to the unusual action of odontoclasts, which resorb dentine from within the pulp of the tooth. Internal tooth resorption is widespread in extant snakes, differs from replacement in other reptiles, and is even detectable via non-destructive μCT scanning, providing a method for identifying fossil snakes. We then detected internal tooth resorption in the fossil snake *Yurlunggur*, and one of the oldest snake fossils, *Portugalophis*, suggesting that it is one of the earliest innovations in Pan-Serpentes, likely preceding limb loss.

Snakes are among the most speciose groups of reptiles, numbering over 3700 extant species[1]. Despite their unique bauplan, the evolutionary relationships of snakes to other lizards have been extensively debated[2–9]. Furthermore, the fossil record of early snakes is fragmentary, represented by isolated or associated bones, which hinders our ability to confidently identify fossil snakes or establish the relative timing of the appearance of key snake features[7,9,10]. However, as limb-reduced, and eventually limbless predators, novel craniodental morphologies were particularly important to the evolutionary success of early and modern snakes[11,12]. We would therefore expect to observe changes in dental anatomy early in their evolutionary history that are present in stem- and crown-group snakes[13]. For this reason, we investigated the development and evolution of a well-known, but poorly understood feature of extant snake dental anatomy, their tooth replacement mode, and compared this with key snake fossils.

One of the most conspicuous features of a snake's dentition is a lack of external resorption pits along the lingual surfaces of the teeth, the absence of which also occurs in *Heloderma*, *Lanthanotus*, and

*Varanus*[3,5,7,14–19]. Snakes can have multiple generations of replacement teeth forming behind each functional tooth, indicating very frequent replacement, but they show no external signs of resorption until an old tooth is about to be shed. External resorption normally signals the onset of tooth replacement, and visible resorption pits do occur in virtually all other toothed amniotes[14,17]. Despite the striking difference between tooth replacement in snakes and other amniotes[17,19,20], we still do not know: (1) how snakes shed their teeth without showing obvious external signs of tooth resorption; (2) if their replacement mode can be further distinguished from those of other lizards; and (3) if it is identifiable in fossil snakes.

Here, we provide the first analysis of the tooth replacement cycle for extant snakes using Tartrate-Resistant Acid Phosphatase (TRAP) staining, a commonly used histochemical stain for osteoclast/odontoclast differentiation and activity[21–24], to determine how snakes replace their teeth in the absence of external tooth resorption. We then compare these observations with histological thin sections and high-resolution computed tomography (μCT) scans of other

[1]Department of Biological Sciences, University of Alberta, Edmonton, AB, Canada. [2]Centre for Oral, Clinical & Translational Sciences, King's College London, London, United Kingdom. [3]School of Biological Sciences, University of Adelaide, Adelaide, SA, Australia. [4]South Australian Museum, Adelaide, SA, Australia. [5]Centre for Craniofacial & Regenerative Biology, King's College London, London, United Kingdom. [6]Instituto de Plasmas e Fusão Nuclear, Instituto Superior Técnico, Universidade de Lisboa, Lisbon, Portugal. [7]CERENA, Instituto Superior Técnico, Universidade de Lisboa, Lisbon, Portugal. ✉e-mail: aaron.leblanc@kcl.ac.uk

alethinophidian and scolecophidian snakes, and to thin sections and skeletal material of several other lizards to identify key differences between the snake tooth replacement cycle and the two other canonical replacement modes in squamates (iguanid- and varanid-type[14]). These comparisons reveal anatomical markers for snake-type tooth replacement that are recognizable with µCT scanning, allowing us to identify snake-type tooth replacement non-destructively. In testing this on the archaic madtsoiid snake *Yurlunggur* from the early Miocene of Australia[25] and one of the oldest snake fossils, *Portugalophis lignites* from the Jurassic of Portugal[7], we show that snakes possess an unusual form of tooth replacement that differs from other reptiles. The presence of this replacement mode in extant alethinophidians, scolecophidians, and the Upper Jurassic snake *Portugalophis* further suggests that it was acquired early in the evolutionary history of snakes, and may even predate limb loss.

## Results

### TRAP staining reveals internal resorption during snake tooth replacement

Our coronal and parasagittal TRAP-stained sections of the corn snake, *Pantherophis guttatus*, revealed populations of mono- and multinucleated TRAP-positive cells at multiple points in the tooth replacement cycle in the dentary, maxillary, and palatal tooth rows (Fig. 1). In each tooth row the cycle was the same. We identified five stages, some of which follow Sahara's[26] staging scheme for the resorption of coronal dentine in mammals: (1) attachment, (2) ankylosis, (3) pre-resorption, (4) resorption, and (5) shedding of old teeth.

During the initial attachment of a new tooth to the tooth-bearing element, the fibrous attachment tissue along the base of a tooth (the periodontal ligament, sensu[27]) was unmineralized and anchored the tooth to a thin layer of alveolar bone forming the new tooth socket (Fig. 1a, e). The pulps of these new teeth were lined by the elongate cell bodies of odontoblasts, the cells responsible for the centripetal production of dentine. Newly attached teeth were not associated with TRAP-positive cells (Fig. 1a, e).

Once all the attachment tissues had mineralized, the tooth was ankylosed to the jaw. At this stage, odontoblasts continued to deposit dentine, thickening the wall of the tooth (Fig. 1b). Most of the teeth at this stage had several TRAP-positive cells within the pulp, including occasional large, multinucleated cells (Fig. 1f). Additionally, some cells within the pulp were smaller and stained weakly positive for TRAP.

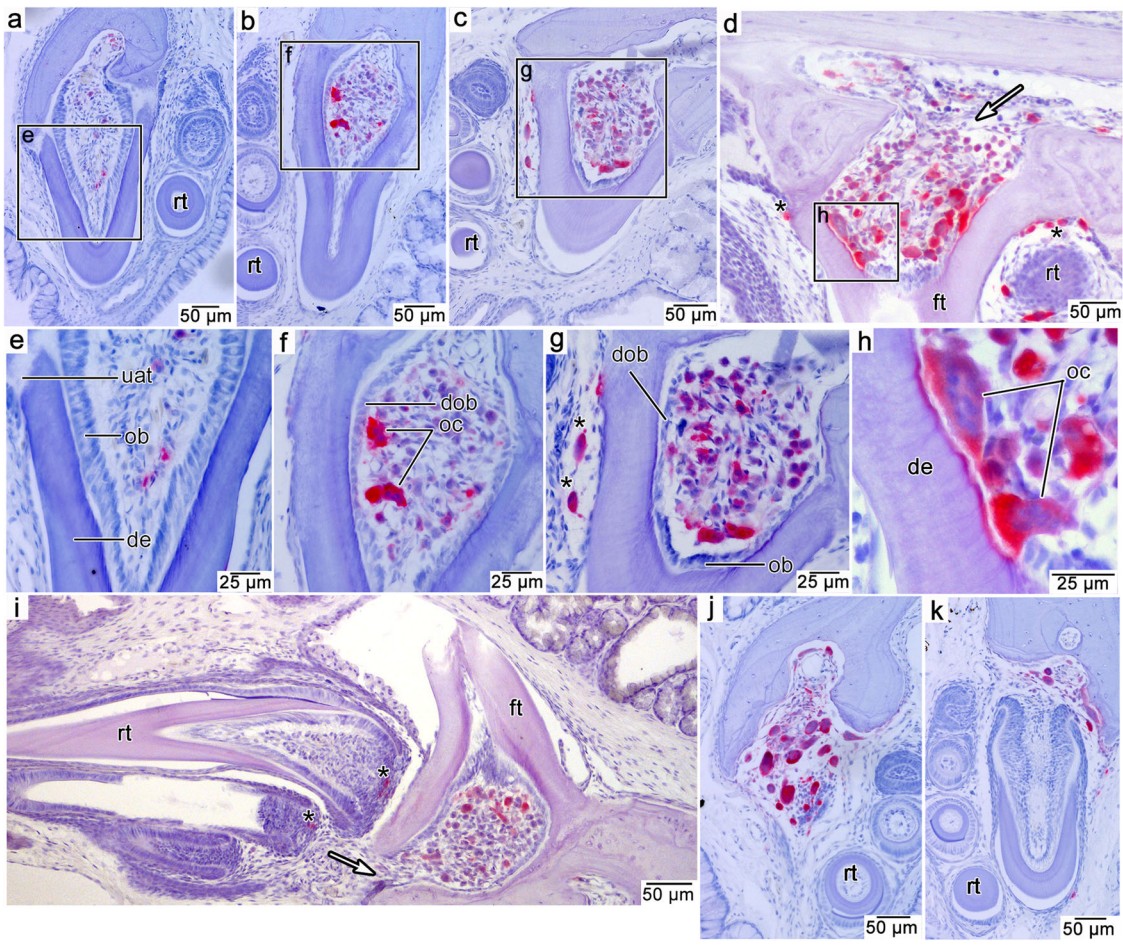

**Fig. 1 | TRAP-stained sections of the tooth replacement cycle in a corn snake (*Pantherophis guttatus*).** TRAP-positive odontoclasts stain red. Sections are counterstained with hematoxylin and eosin. **a** Attachment stage in a palatal tooth in coronal section. **b** Ankylosis stage in a maxillary tooth. **c** Pre-resorption stage in a maxillary tooth. **d** Early resorption stage in a palatal tooth in parasagittal section. White arrow indicates a potential entry point for odontoclast precursors. **e** Closeup of the dental pulp in a showing columnar odontoblasts and absence of odontoclasts. **f** Closeup of dental pulp in b showing shorter, degenerating odontoblasts and multinucleated odontoclasts. **g** Closeup of dental pulp in c showing degenerating odontoblasts and large multinucleated odontoclasts and smaller mononucleated odontoclasts. **h** Closeup of a smaller region in the dental pulp in d showing large, multinucleated odontoclasts resorbing the pulpal surface of the dentine. **i** Pre-resorption stage in a dentary tooth in parasagittal section. White arrow indicates potential entry point for odontoclast precursors. **j** Shedding stage in a palatal tooth. **k** Replacement stage in a maxillary tooth. de dentine, dob degenerating odontoblast, ft functional tooth, oc odontoclast, rt replacement tooth. Asterisks indicate odontoclasts not associated with pulpal resorption.

These resemble early-stage, or pre-odontoclasts[23,26]. The odontoblasts still lined the inner margins of the dentine. However, odontoblasts were no longer columnar in shape. These flatter cell bodies resembled the degenerated odontoblasts that occur in mammalian teeth prior to their displacement by odontoclasts along the inner walls of the dentine[26]. There was no evidence of TRAP-positive cells along the outside of the teeth.

At the pre-resorption stage, columnar odontoblasts occasionally remained in crownward regions of the pulp whereas they were reduced to flat cell bodies in more basal regions (Fig. 1c, g). The remaining portion of the pulp was filled with different sizes of round TRAP-positive cells that appeared to increase in size towards the crown end of the pulp (Fig. 1g). This same phenomenon occurs in the late pre-resorption stage in the coronal pulp of human teeth, prior to tooth shedding[26,28]. At this stage, small numbers of odontoclasts occasionally formed along the outside of the tooth in association with the approaching replacement tooth, but they did not form a detectable resorption pit (Fig. 1g).

The subsequent resorption stage is apparently a short-lived phase in the tooth replacement cycle of *P. guttatus*. We only observed the resorption stage in a single tooth in serial sections through nearly all the tooth rows in a single head. We did observe this stage in several other snake species, but not with TRAP staining (see below). At this point, the odontoblasts had been replaced by large, mostly multinucleated TRAP-positive odontoclasts (Fig. 1d, h). These odontoclasts resorbed the dentine of the tooth from the inside and did not seem to show a preference for one side of the tooth or the other. Instead, the tooth root and the base of the crown were resorbed centrifugally from the inside. We observed TRAP-positive cells along the outside of a resorption-staged tooth, but these cells were smaller than those found inside of the pulp and still did not form any detectable resorption pits (Fig. 1i).

Tooth shedding in snakes occurs when the base of a tooth has been sufficiently resorbed internally to allow it to break away from the jaw. While we did not observe the moment of tooth shedding in *P. guttatus*, we did observe this stage in other snakes (see below). Several tooth positions in this *P. guttatus* specimen were large resorptive crypts devoid of attached teeth. These crypts were often filled with large odontoclasts that were resorbing the remaining alveolar bone (Fig. 1j). These cells were probably associated with the resorption and shedding of the previous tooth generation, and they continued to resorb the surrounding tissues to accommodate the approaching replacement tooth (Fig. 1k).

## Internal tooth resorption is widespread in modern snakes

To determine if this replacement mode in *P. guttatus* is representative of snake tooth replacement in general, we compared it with histological and µCT data from other species occupying key positions in snake phylogeny. We found the same tooth replacement mode in thin sections and µCT scans of several other modern snakes (*Hydrophis cyanocinctus, Oxyuranus scutellatus, Acrochordus javanicus, Crotalus atrox, Boiga dendrophila, Boa constrictor,* and *Malayopython reticulatus*) (Figs. 2–4). These comparisons also allowed us to fill in the missing phases of the replacement cycle that we did not observe in *P. guttatus*.

In all snake taxa we examined, we did not find any notable evidence of external resorption until the end of the tooth replacement cycle when a tooth was about to be shed (Figs. 2, 3; Supplementary Figs. 1–5). In our histology sections of the sea snake *Hydrophis cyanocinctus*, we identified early and late-stage resorption by the presence of large multinucleated odontoclasts lining the pulpal side of the dentine, resulting in scalloped internal surfaces (Howship's lacunae) in dentary, maxillary, and palatine teeth (Fig. 2). As in *P. guttatus*, at the earliest stages of resorption in *H. cyanocinctus*, the odontoclasts replaced the odontoblast layer and began resorbing dentine from the inside of the pulp chamber outwards. This process

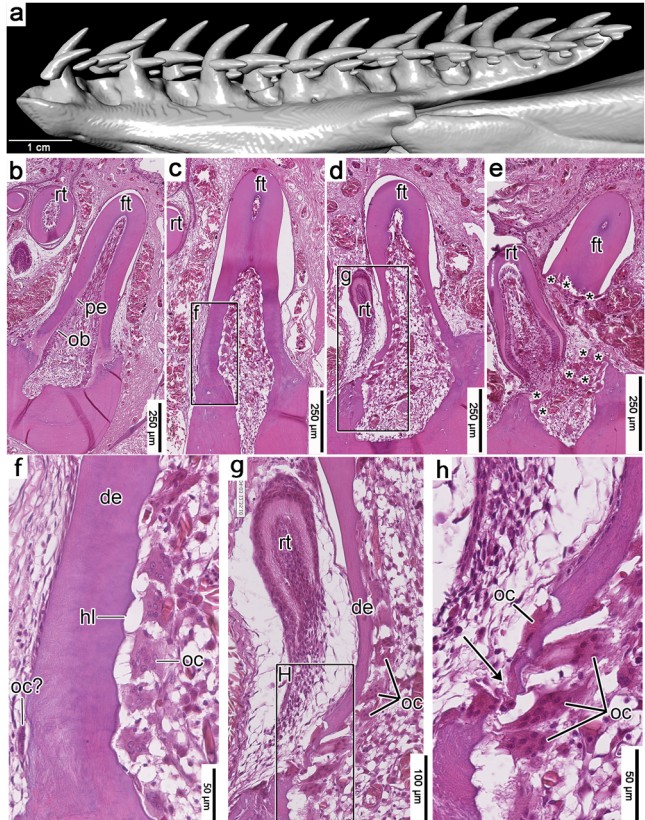

**Fig. 2 | Histology of snake-type tooth replacement in the sea snake *Hydrophis cyanocinctus* (hematoxylin and eosin staining). a** CT surface rendering of a dentary of *H. cyanocinctus* (SAMA R70272) in lingual view. **b** Recently attached dentary tooth in coronal section. Note the lining of odontoblasts and lighter-colored predentine along the inner walls of the tooth. **c** Tooth resorption inside a functional maxillary tooth (flipped for comparisons). **d** Late-stage tooth resorption and initial pulp breach from within a dentary tooth. **e** Tooth shedding and near simultaneous tooth replacement in a palatine tooth (flipped vertically for consistency). **f** Closeup of lingual dentine wall in c showing smooth, nearly unresorbed external surface and a scalloped, partially resorbed, and odontoclast-lined internal surface. **g** Closeup of lingual surface of functional tooth and replacement tooth in d showing extensive internal resorption and a very small pulp breach along the base of the functional tooth. **h** Closeup of the small pulp breach in g showing a small population of odontoclasts to the external surface of the tooth base. de dentine, ft functional tooth, hl Howship's lacuna, ob odontoblasts, oc odontoclasts, pe predentine, rt replacement tooth. Arrowhead indicates position of pulp breach in (H). Asterisks in (E) mark odontoclasts.

continued until the tooth base was so thin that small breaches of the dentine wall began to form approximately at the same time as the replacement tooth moved into position (Fig. 2d, g, h). At this stage, odontoclasts lined the inside of the tooth and the breached areas of the functional tooth base. Nearly all the resorption occurred inside of the functional tooth (Fig. 2c), but a few odontoclasts were found on the outside of the pulp chamber at the latest stages of tooth resorption (Fig. 2h).

Resorption continued until the entire functional tooth separated from the jaws (Fig. 2e). However, for at least a brief time in the replacement cycle, the old detached tooth can remain in the mouth, suspended by gingival tissues as the replacement tooth continued its advance. As we observed in *P. guttatus*, odontoclasts were scattered around the growing crypt and continued to resorb portions of dentine and bone as the replacement tooth moved into position. We found the same internal tooth resorption mechanism and pulpal odontoclasts in histological sections of another colubroid, the coastal taipan *Oxyuranus scutellatus* (Supplementary Fig. 1a, b), and in more distantly

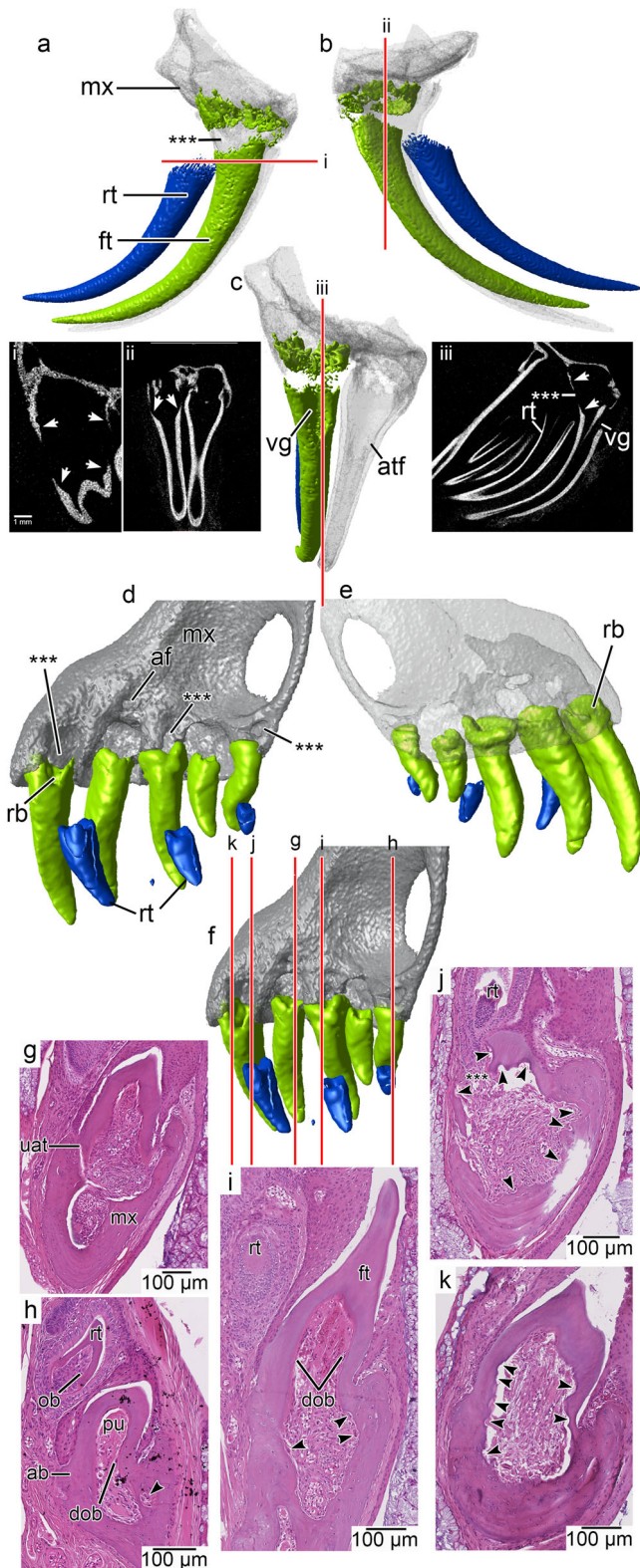

**Fig. 3 | Tooth resorption in the venom fang of the viperid *Crotalus atrox* and in the scolecophidian snake *Anilios bicolor*. a** CT-segmented model of a maxilla of *Crotalus atrox* showing the near-shedding stage of a venom fang in lateral, **b** lingual, and **c** anterior views. The old functional fang is in green, the replacement fang is in blue. CT slices in various angles show how internal resorption has allowed for the controlled detachment of the old fang from its base (i–iii). The neighboring functional fang is undisturbed in the process. **d** CT-segmented model of a maxilla of *Anilios bicolor* showing teeth at various stages in the replacement cycle in lingual (functional teeth in green, replacement teeth in blue); **e** labial, and **f** posterolingual views to visualize large resorption structures in the maxilla (asterisks) and their relationships to the teeth. **g** Coronal section of a new tooth that is attached to the maxilla by non-mineralized attachment tissues. **h** Ankylosed tooth in which the internal portion of the base of the tooth has begun showing signs of resorption (arrow) while the replacement tooth is still a distance away from the functional tooth base. Note the well-developed odontoblasts in the replacement tooth compared with the degenerated odontoblasts within the functional tooth. **i** Ankylosed tooth showing appearance of larger macrophages and early signs of resorption within the maxilla (arrows). The new replacement tooth does not initiate resorption on the outer, lingual side of the functional tooth, similar to other snakes. **j** Late resorption-staged tooth with a pulp breach on its lingual side (asterisks). The pulp shows signs of resorption in all directions and an abundance of odontoclasts (arrows). **k** Same tooth as in j showing the extent of resorption within the pulp. All thin sections are hematoxylin and eosin stained and flipped vertically for clarity. ab alveolar bone, atf attached fang, dob degenerated odontoblasts, ft functional tooth, mx maxilla, ob odontoblasts, pu pulp, rb resorbed tooth base, rt replacement tooth, uat unmineralized attachment tissue, vg venom groove.

we predicted that the early- and late-stage resorption stages would be identifiable by the Howship's lacunae that develop along the internal walls of the teeth. Our μCT scans were of sufficient resolution in most cases to identify teeth in the process of being replaced. Newly added teeth had smooth internal walls and showed no evidence of Howship's lacunae internally (Supplementary Figs. 3 and 5; Supplementary Movies 1 and 4). As in our thin sectioned samples, partially resorbed teeth have intact outer walls, but have excavated internal pulp chambers, as well as visible Howship's lacunae in more apical regions of the pulp (Figs. 3 and 4; Supplementary Figs. 3, 5–7; Supplementary Movies 2, 3, 5). This is also evident in the venom fangs of the viperid *Crotalus atrox* (Fig. 3a–c). Here, internal resorption caused the complete detachment of a fang from its base prior to the arrival of the replacement tooth to its final position. This allows for precise removal of the old venom fang without causing any collateral resorption to the neighboring functional fang, which would still be connected to the venom gland.

## The scolecophidian snake *Anilios bicolor* shows an internal resorption mechanism

Scolecophidians have a contended phylogenetic position as either the earliest diverging snake lineage(s)[4,15,29], or a more highly nested group relative to several fossil taxa[2,9,25,30]. Determining the replacement mode in scolecophidians is therefore important for future phylogenetic reconstructions of stem and crown Serpentes (i.e., Pan-Serpentes[31], a clade sometimes referred to as Ophidia[7,30,32]). However, many members of this group have extremely small or reduced dentitions (e.g., Leptotyphlopidae), or are endangered (e.g., Anomalepididae), which hinders destructive histological analysis. Fortunately, we were able to scan and section one of the more common and larger species, the Australian typhlopoid *Anilios bicolor*, thus providing the first in-depth analysis of tooth replacement in a scolecophidian snake.

Our preliminary μCT data did not reveal any signs of internal resorption, because the teeth were too small, and the scan resolution was limited. The initial scan showed resorption pit–like holes forming along the bases of some of the maxillary teeth in *A. bicolor* (Fig. 3d, f). These pits were not homologous with the alveolar foramina, which were separate, smaller holes in the maxilla (Fig. 3d, contra[17]). However,

related snakes, *Acrochordus javanicus* and *Boa constrictor* (Supplementary Figs. 1c, d and 2).

We also examined μCT scans of the same *B. constrictor* maxilla prior to sectioning, a maxilla of *Malayopython reticulatus*, a maxilla of the viperid *Crotalus atrox*, the skull of another colubrid (*Boiga dendrophila*), and a skull of the scolecophidian *Anilios bicolor* to determine if this internal resorption mechanism is detectable by non-destructive means. While the pre-resorption stage would be impossible to detect,

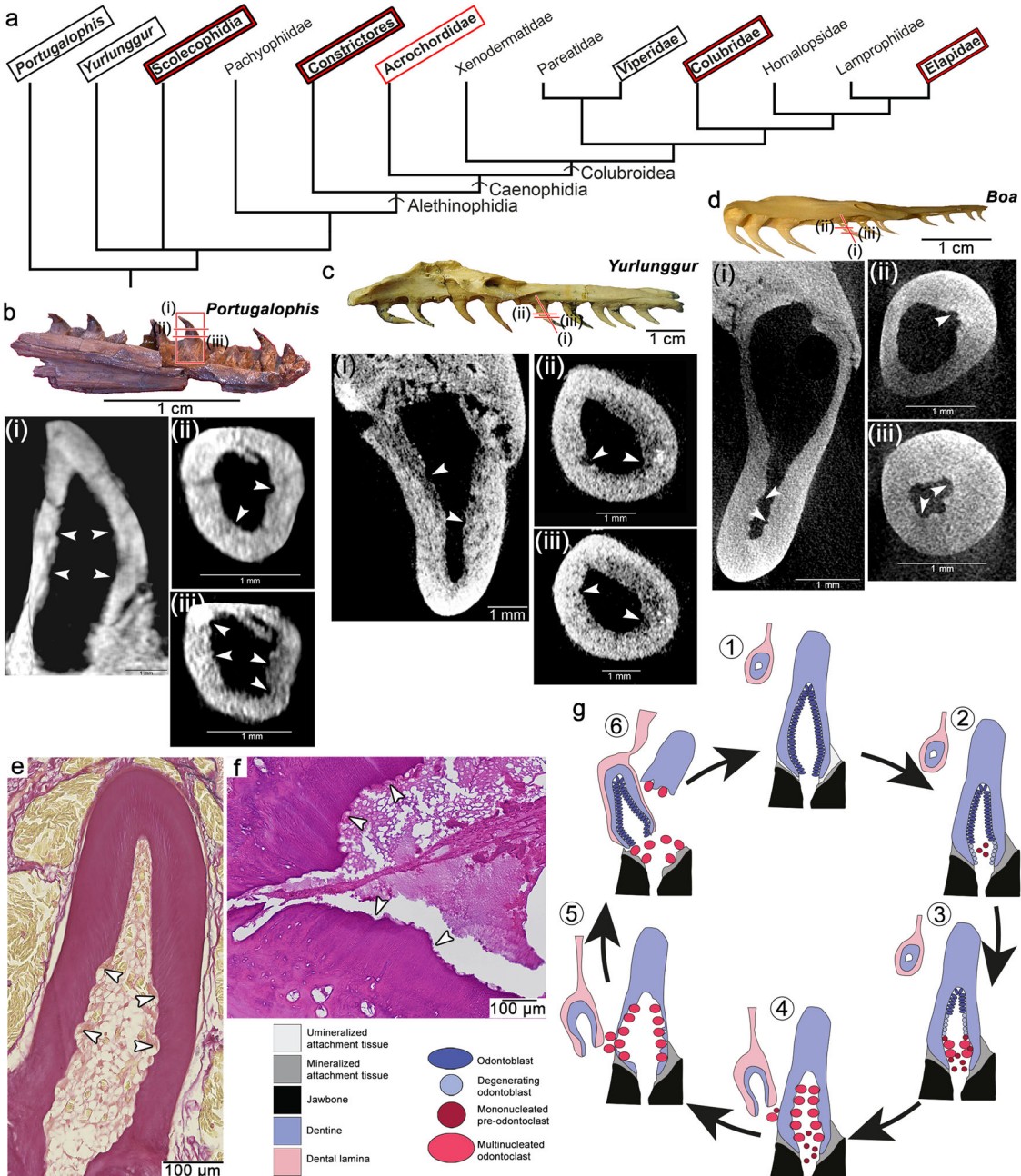

**Fig. 4 | Observed occurrences of internal tooth resorption in extant and fossil snakes. a** Abbreviated phylogeny of Pan-Serpentes, showing relative placement of modern and fossil snakes sampled in this study (phylogeny modified from[7,12,57]). Bolded taxon names indicate occurrences of internal tooth resorption observed with histology sections (red boxes) or µCT scanning (black boxes). **b** Paratype dentary of *Portugalophis lignites* (MG-LNEG 28094) from the Jurassic of Portugal showing internal resorption in one of the teeth in virtual sagittal (i) and serial transverse (ii, iii) sections. **c** Maxilla of *Yurlunggur* sp. [Queensland Museum, Brisbane, Australia (QMF) 45391] from the Miocene of Australia[58] showing internal

resorption in one of the teeth in virtual frontal (i) and serial transverse (ii, iii) sections. **d** Maxilla of a modern *Boa constrictor* showing internal resorption in one of the teeth in virtual frontal (i) and serial transverse (ii, iii) sections. **e** Histological frontal section through a palatine tooth of *Hydrophis cyanocinctus* showing internal dentine scalloping caused by odontoclasts. **f** Closeup of internal dentine scalloping in a histological transverse section through a *Boa constrictor* maxilla. **g** Model of the snake tooth replacement cycle from (1) attachment, (2) ankylosis, (3) pre-resorption, (4) early resorption, (5) late resorption, and (6) shedding stages. White arrowheads indicate Howship's lacunae.

differing from classic resorption pits in other tetrapods (see below), the larger pits in *A. bicolor* first appeared on the bone of the maxilla and well in advance of the encroaching replacement tooth (Fig. 3d). Furthermore, histological sections showed that extensive resorption occurred from within the maxilla, below the pulps of the functional teeth (Fig. 3h, i; Supplementary Fig. 4). At the same time, replacement teeth formed high along the jaw margins, well away from the tooth bases, and approached the bases of the functional teeth by elongation

of dental tissues rather than migration of the tooth bud towards its predecessor, the latter of which occurs in non-snake squamates (Fig. 3d; see below).

By the time the soft tissues of the replacement tooth base had extended close enough to the functional tooth to cause external resorption, the inner walls of the pulp chamber of the functional tooth were already scalloped in all directions (Fig. 3j, k), and had even detached the tooth from the maxilla along its labial edge

(Supplementary Fig. 4f). Moreover, we found that the massive amounts of resorption occurring deep within the maxilla were continuous with the pulps of each tooth and with the foramina we first detected along the bone of the maxilla (Supplementary Fig. 4).

The significance of this extensive internal resorption in the maxilla is unclear; however, the abundance of Howship's lacunae and odontoclasts within the pulps of some of the teeth indicate that internal resorption preceded the formation of externally visible breaches on the lingual surfaces of the tooth bases (Supplementary Fig. 4e, f). Our histological data indicate that the teeth of *A. bicolor* passed through the same stages as other snakes described above: replacement teeth do not initiate resorption, the odontoblasts inside functional teeth degenerate, and the pulp cavity soon becomes a resorptive front inside the tooth (Fig. 3g–k).

### Internal resorption is detectable in fossil snakes

Based on the success of μCT scanning, we predicted that internal resorption would also be detectable in the fossil record, thus providing a powerful tool for identifying isolated jaws of crown- and even stem-snake fossils. Unfortunately, several key fossils are either edentulous (due to post-mortem decay of their ligamentous tooth attachment[9,33]), have broken teeth[7], or occur on large slabs[2,4], hindering conventional high-resolution CT imaging. However, disarticulated tooth-bearing jaws of two important snake fossils were available for μCT scanning: a maxilla from the extinct basal snake *Yurlunggur* (Madtsoiidae) from the early Miocene of Australia[25], and a dentary of one of the earliest stem snakes, *Portugalophis lignites* from the late Jurassic of Portugal (Fig. 4; Supplementary Fig. 7). These scans revealed similar evidence of Howship's lacunae within the teeth that were undergoing early-stage resorption. The resorption structures are scallop-shaped indentations visible in multiple digital sections of teeth in the early stages of resorption (Fig. 4b, c; Supplementary Figs. 5k–o, 7; Supplementary Movies 6–8). These structures are identical to those we identified in μCT scans of *Malayopython* and *Boa* (Fig. 4d–f; Supplementary Fig. 5a–j). We found no evidence of resorption along the external walls of the teeth in either specimen, further indicating that these ancient snakes possessed an internal form of tooth resorption, like their modern relatives.

### Snake-type tooth replacement differs from that of iguanids and varanids

Unlike snakes, the earliest stage of the iguanid-type tooth replacement cycle is the resorption of the external lingual surface of a tooth, which begins along the cementum coating the tooth root and then the outer dentine layers (Fig. 5a, b). External resorption of the functional tooth can begin early in replacement tooth development when the epithelial cells of the replacement tooth bud are organized into discrete layers but have not yet formed any dentine and enamel (Supplementary Fig. 8). We identified Howship's lacunae along the lingual surfaces of the tooth roots in the iguanids *Iguana* and *Sauromalus*, and the teiid *Aspidoscelis* (Fig. 5b–d; Supplementary Fig. 8). At early stages in the replacement cycle, the pulp showed no evidence of internal resorption and was still lined with dentine-producing odontoblasts (Fig. 5b). External resorption continued until the resorption pit reached the pulp, allowing odontoclasts to invade through the newly formed opening in the dentine (Fig. 5c; Supplementary Fig. 8). This differs from the condition in alethinophidian and scolecophidian snakes, where the pulp had already been transformed into a resorptive front by this stage.

As the resorption pit enlarged, the odontoclasts disrupted the odontoblast layer in the pulp (Fig. 5d; Supplementary Fig. 9a). Resorption of the functional tooth then proceeded in a wave, with the lingual pole of the tooth being at a more advanced stage of resorption than the labial pole (Supplementary Fig. 9). At the most advanced stage of tooth resorption, the tooth base was nearly completely resorbed.

Functional teeth are shed at this stage in the replacement cycle because they no longer have any attachment to the jaws or the neighboring teeth. The replacement tooth eventually occupies the position of the shed tooth and erupts into the same position.

In *Varanus*, *Heloderma*, and *Lanthanotus* the replacement teeth form posterolingual to their functional teeth, similar to snakes[14,17,20,34], and without obvious signs of resorption pits[14]. However, unlike snakes and more like the iguanid-mode of tooth replacement, the initial phase of resorption in varanid-type tooth replacement still begins along the external surface of the functional tooth, along its lingual surface, and in proximity to the approaching replacement tooth (Fig. 4e–i). Although a macroscopic resorption pit is absent, odontoclasts still form Howship's lacunae along the external, lingual surface of the functional tooth, giving it a scalloped appearance (Fig. 5i) that has also been observed via electron microscopy[34]. We also identified external signs of tooth resorption in *Varanus* based on osteological specimens (Fig. 5f, g; Supplementary Fig. 10). These clearly show that tooth resorption is detectable externally at early stages in the replacement cycle, despite the lack of obvious resorption pits. Unlike in snakes, our thin sections did not reveal any evidence of internal (pulpal) Howship's lacunae at this stage in the replacement cycle in *Varanus* (Fig. 5h, i).

At later stages of tooth resorption, the pulp chamber of the functional tooth was breached, but along a different surface than that of iguanid-type teeth. We identified a resorptive breach of the pulp chamber along the distal (posterior) surface of the functional teeth (Fig. 5j). This breach is produced by large odontoclasts that then migrate into the honeycomb-like pulp of the tooth base (the extensive plicidentine at the base of the teeth in *Varanus* partitions the pulp into a series of smaller chambers[34,35]). At this stage, odontoclasts resorbed large portions of the tooth base. This stage is also visible in osteological specimens, where the tooth base has been resorbed from all sides, advancing up the tooth to the base of the crown, undercutting it, and eventually causing it to be shed (Fig. 5f, g; Supplementary Fig. 10). This mechanism would also explain similar undercutting of functional teeth in *Heloderma* (Supplementary Fig. 11).

## Discussion

### Snakes have evolved a distinct tooth replacement mode

The snake sample presented here shows that they are the outliers in squamate tooth replacement, and tetrapod tooth replacement more generally[14,36]. Nearly all amniotes (mammals, crocodilians, and other squamates) show evidence of external tooth resorption, leaving a visible pit in the dentine base associated with the formation of a new tooth, which eventually breaches the pulp chamber of its predecessor[14,20,34,37–42]. Tooth resorption in most reptiles, including iguanids and varanids, occurs in close association with the encroaching replacement tooth, whereas it radiates in all directions away from the pulp within a snake tooth. The conventional models of tooth resorption and replacement rely heavily on mammalian experimental studies[23,26,28,43–45], but these provide clues for interpreting the evolution of internal tooth resorption in snakes. A prevailing hypothesis is that the pressures of an approaching replacement tooth initiate odontoclastic activity[45,46], which leads to external root resorption. An external layer of odontoclasts, which in mammals are recruited by cells from the replacement tooth or the periodontal tissues of the functional tooth[23,39,44,47], is also found in iguanid and even varanid-type tooth replacement modes (Fig. 5), supporting this general model. However, there is no association between a replacement tooth and odontoclast recruitment in snakes. The onset of resorption is independent of the proximity of the replacement tooth to the functional one (Fig. 1)[48], even in scolecophidian snakes (Fig. 3d–k).

In snakes, virtually all the odontoclastic activity instead occurs within the pulp of a functional tooth, with only a small number of odontoclasts appearing externally in association with the advancing replacement tooth (Figs. 1–3). Odontoclasts enter the pulp chamber

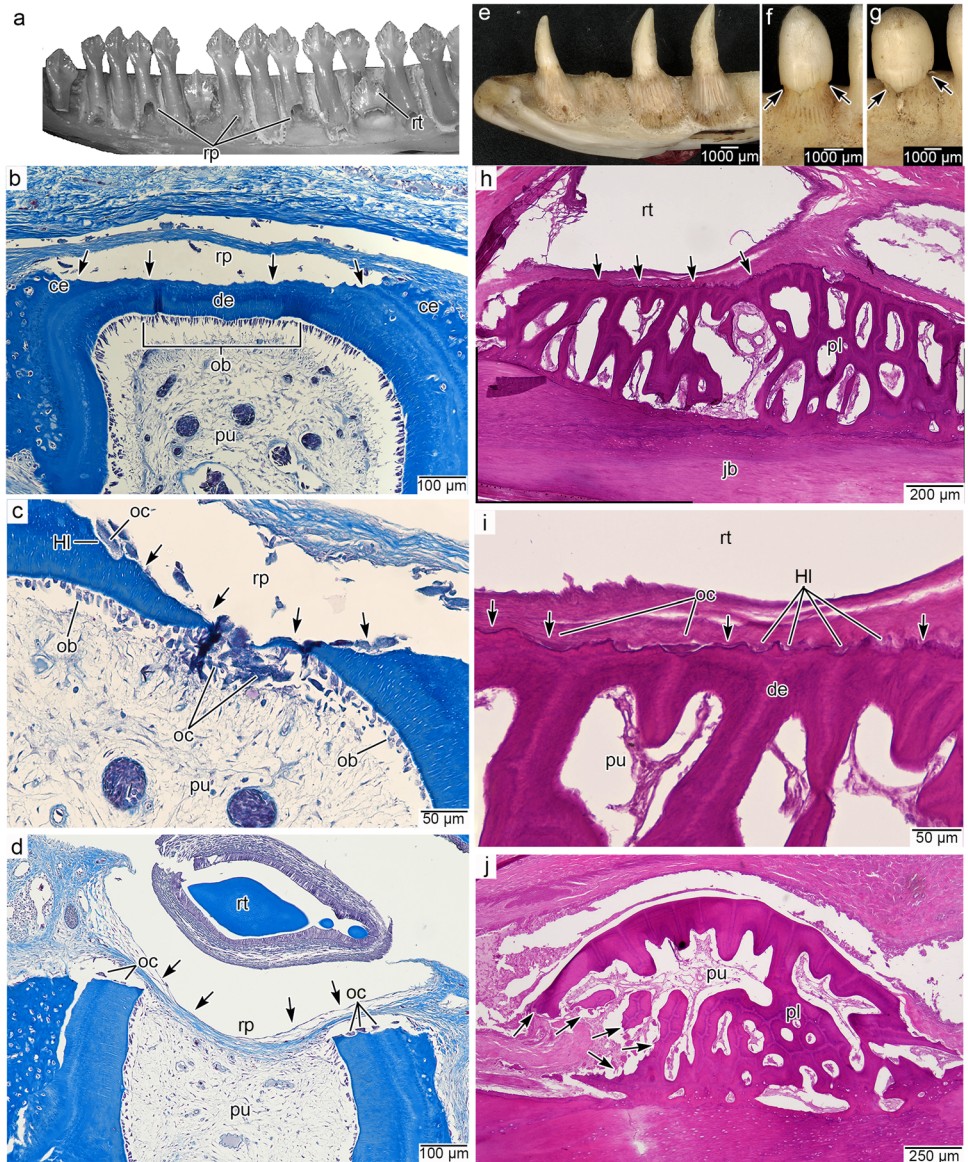

**Fig. 5 | Iguanid- and varanid-type tooth resorption in *Iguana* and *Varanus*.**
**a** Lingual view of a dentary of *Iguana delicatissima* (MCZ 10975) exhibiting typical resorption pits. Credit: Museum of Comparative Zoology, Harvard University ©President and Fellows of Harvard College. **b** Transverse section of a tooth root in an *I. iguana* dentary (Masson's trichrome staining) with a developing resorption pit along its lingual surface. Lingual is towards the top. **c** Transverse section through an *I. iguana* dentary tooth with a replacement pit that has just breached the pulp (Masson's trichrome). Lingual is towards the top. **d** Transverse section through an *I. iguana* dentary tooth with a large replacement pit that has fully breached the pulp (Masson's trichrome). Lingual is towards the top. **e** Lingual view of a right dentary of *Varanus* sp. showing a lack of resorption pits. (MoLS X265). **f** Lingual view of a dentary tooth of *Varanus niloticus* (MoLS X266) showing an advanced stage of external tooth resorption, with no signs of a resorption pit. **g** Lingual view of another *V. niloticus* dentary tooth at a more advanced stage of tooth resorption, with an undercut tooth crown. **h** Transverse section of a functional tooth showing the earliest stages of external tooth resorption by an encroaching replacement tooth (hematoxylin and eosin). Lingual is towards the top. **i** Closeup of the external resorption surface in **h** showing Howship's lacunae and presumed odontoclasts (hematoxylin and eosin). **j** Transverse section of a functional tooth showing an early pulp breach by odontoclasts along its posterior wall (hematoxylin and eosin staining). Lingual is towards the top. Arrows indicate directions of resorption.

without forming an external breach of the tooth surface. They instead enter through the alveolar vascular network, or differentiate from progenitor cells within the pulp, as they do in late stages of tooth resorption in mammals[26,28]. In mammals, a population of odontoclasts appears in the dental pulp first as mononucleated precursor cells, but only after the root has been resorbed externally[26]. In snakes, this re-organization of the dental pulp occurs much earlier, is independent of external root resorption, and occurs shortly after a new tooth is fully attached to the jaws, before a replacement tooth is fully formed. These comparisons suggest that snake-type tooth replacement may have resulted from an evolutionary shift in the timing of these two processes. The initial stage of external resorption has been nearly completely suppressed (though limited external odontoclastic activity remains; Fig. 1), leading to an absence of true resorption pits, and the stereotypically late stage of pulpal resorption has become the dominant resorptive mechanism.

## Functional advantage of snake-type tooth replacement

Teeth are crucial for limb-reduced or limbless predators, given that these become essential for prey capture and transport into the mouth[49]. Internal tooth resorption characterizes marginal and palatal teeth in snakes (Figs. 1 and 2), but is absent in other limbless squamates. This replacement mode could impart a selective advantage by maintaining a continual supply of functional teeth for early snakes with

limb-reduced bodies and may be correlated with wholesale ingestion of large prey (macrophagy) in early limb-reduced and eventually limbless predators. The snake tooth replacement mode contracts a key phase of the tooth replacement cycle by suppressing external resorption and minimizing the duration of the edentulous stage between when an old tooth has been shed and a new tooth has yet to attach to the jaws (Figs. 2e and 3). Snakes can therefore replace their teeth comparatively quickly with fully developed successors, with a recent estimate suggesting that the complete tooth replacement cycle in a snake can last only 3 weeks[49].

This is an important adaptation in animals that frequently lose several of their sharp, brittle, shallowly implanted teeth while attempting to ingest relatively large prey items. For example, in a radiograph of a specimen of *Python curtus* we counted almost a dozen teeth lost in a single meal (Supplementary Fig. 12). This replacement mode is also essential for rapidly replacing the fangs in venomous species, where the need to maintain functional teeth is crucial for survival (Fig. 3a–c). Moreover, internal resorption also limits collateral resorption in many species: scalloping of the bases of neighboring teeth that prematurely weakens them before they are ready to be replaced. This is a rare phenomenon in snakes, but happens frequently in other squamate taxa[50] (Supplementary Fig. 13).

## Internal tooth resorption is an important snake synapomorphy

Snake skulls are highly modified compared to other lizards, creating significant challenges and disagreements in estimating the phylogenetic relationships of snakes among Squamata using anatomical characters[5,7,9,11,13,29,30,32,51]. Cranial remains of Cretaceous and, more recently, Jurassic snakes have revealed a mosaic of snake- and lizard-like morphologies, thus increasing the need to identify anatomical synapomorphies of stem- and crown-group snakes. Extinct madtsoiid snakes such as *Yurlunggur* are widely considered as either basal to all living snakes (scolecophidians and alethinophidians)[7–9,25,52] or a sister taxon to all extant alethinophidian snakes[10,13,30], and the Jurassic-aged *Portugalophis* is part of the oldest snake lineage yet known[7,30]; the observation that both show internal tooth resorption similar to modern snakes (Fig. 5), and the fact that there are no known examples of fossil snakes with typical external resorption pits[7,13,17] suggest that this tooth replacement mode predates the evolutionary divergence of scolecophidians and alethinophidians and is a shared feature of Pan-Serpentes.

Our results showcase how high-resolution, non-invasive μCT scans could be used to support or refute the identifications of partial cranial remains of other fossil taxa using the internal tooth resorption mechanism we describe here. Even scolecophidian snakes, despite their unusual anatomy, lack true resorption pits[17,53] and follow a modified version of the internal resorption mechanism seen in other fossil and extant snakes (Fig. 3d–k). Previously, amphisbaenians were also thought to lack resorption pits[15], however, personal observation of large resorption pits associated with replacement teeth in *Amphisbaena* [e.g., British Museum of Natural History, London, UK (BMNH) 1964.1811; Supplementary Fig. 14] suggests they too replace their teeth similar to other non-snake squamates. Furthermore, the absence of this replacement mode in the closest snake outgroups (Anguiformes and Iguania, sensu[54]), leads us to conclude that internal tooth resorption is a synapomorphy of Pan-Serpentes.

Internal resorption can thus join the features characterizing snakes and can be used to identify early members of the group. *Yurlunggur* and other madstoiids have been recovered as a basal snake lineage that predates the divergence of the two major groups of extant snakes (scolecophidia and alethinophidia) by several[8,9,25] (though not all[13,30]) phylogenetic analyses, and *Portugalophis* predates the known loss of forelimbs in snakes by about 50 million years[2,7,52]. Snake-type tooth replacement is therefore a promising feature for identifying putative and even fragmentary snake fossils from the key periods of snake evolutionary history.

## Methods

This research was conducted with all necessary permissions for destructive sampling of museum specimens.

### Histology

Serial thin sections of several preserved squamate specimens were made for this analysis (Supplementary Data 1). Heads of a juvenile corn snake (*Pantherophis guttatus*) and an adult specimen of scolecophidian snake (*Anilios bicolor*) were decalcified, and then sectioned in transverse and parasagittal sections. The *P. guttatus* individual was cared for and culled at King's College London (KCL) according to UK Home Office approved Schedule 1 methods and regulations in accordance with those set out under the United Kingdom Animals (Scientific Procedures) Act 1986, the European Union Directive 2010/63/EU, and the Amendment Regulations 2012. *A. bicolor* was a cataloged museum specimen [South Australian Museum, Adelaide (SAMA) R1065]. *P. guttatus* sections were examined for osteoclast activity using TRAP staining (Supplementary Information).

The remaining snake and lizard specimens were preserved samples from various institutions or previously made thin sections (Supplementary Data 1). These sections were made at the Advanced Microscopy Facility in the Department of Biological Sciences, University of Alberta and at Histology Services, Department of Health Sciences University of Adelaide. Dissected jaws from *Iguana*, *Varanus*, *Boa*, and *Anilios bicolor* were first decalcified in either Richard-Allen Scientific™ CalRite™ decalcifying and fixative solution (formic acid and formaldehyde) or 10% EDTA for several weeks (with daily solution changes), or in Shandon™ TBD-1™ decalcifier (hydrochloric acid) for several hours. Decalcified specimens were then placed in a dehydration series of toluene and ethanol overnight before being embedded in paraffin wax. Specimens were then sectioned at 5–8 μm thicknesses using a microtome and stained using either Haematoxylin and Eosin or Masson's Trichrome (Supplemental Information).

We also examined additional sections of the snakes *Hydrophis cyanocinctus* (blue-banded sea snake), *Oxyuranus scutellatus* (common taipan) and *Acrochordus javanicus* (Javan file snake), and of the lizards *Aspidoscelis exsanguis* (Chihuahuan spotted whiptail), *Sauromalus ater* (chuckwalla), *Scincus scincus* (common skink), and *Cordylus cordylus* (Cape girdled lizard) that were made for previous studies[12,55,56].

Thin sections of *Pantherophis guttatus* and images of Museum of Life Sciences, King's College London, United Kingdom (MoLS) specimens were done using a Keyence VHX-7000 digital microscope in the Faculty of Dentistry, Oral & Craniofacial Sciences at KCL. Sections of *Iguana iguana*, *Varanus* sp., *Boa constrictor*, *Acrochordus javanicus*, *Aspidoscelis exsanguis*, and *Sauromalus ater* were imaged using a Nikon DS-Fi3 camera mounted to a Nikon Eclipse E600 polarizing microscope and NIS-Elements imaging software. High-resolution images of the histology sections of *Anilios bicolor*, *Hydrophis cyanocinctus* and *Oxyuranus scutellatus* were taken with a NanoZoomer 2.0HT digital slide scanner (Hamamatsu Photonics) at the Faculty of Health and Medical Sciences of the University of Adelaide and visualized in the free software NDP view v.2 (Hamamatsu Photonics).

### μCT scanning

The right maxilla from the same freshly dissected specimen of *Boa constrictor* (UAMZ unregistered specimen) was scanned using a Skyscan 1176 (Bruker) μCT scanner at the Department of Earth Sciences of the University of Alberta (Edmonton, Canada). The main acquisition settings were set to 885 ms for exposure, 100 kV for voltage, 100 μA for current, and 7.84 μm for resolution (filter: Al 0.5 mm). The maxilla of a *Malayopython reticulatus* (SAMA R 27307) was scanned using a Skyscan 1276 (Bruker) μCT scanner at Adelaide Microscopy, University of

Adelaide (Adelaide, South Australia). The settings for this specimen were set to 800 ms exposure, 95 kV for voltage, 200 µA for current, and 10.04 µm for resolution (filter: Al 1 mm). The µCT scan of the scolecophidian *Anilios bicolor* was acquired at the same facility using the same µCT-scanner, with 800 ms for exposure, 100 kV for voltage, 200 µA for current, and 4.04 µm for resolution (filter: Al 0.5 mm). µCT scans of a viperid (*Crotalus atrox*) and a colubrid (*Boiga dendrophila*) head were made in the Center for Craniofacial Regenerative Biology (KCL) using a Scanco MicroCT 50 (300–350 ms exposure, 70 kV, 140 µA, 4.4–7.4 µm resolution).

Pre-existing scans from two additional specimens, *Hydrophis cyanocinctus* (SAMA R70272), and the fossil snake *Yurlunggur* sp. (QM F45391), were used to illustrate different patterns of tooth replacement and verify the presence of internal tooth resorption. Both of these specimens had been scanned at Adelaide Microscopy using a Skyscan 1076 (Bruker) µCT scanner. The settings used in the scan of *H. cyanocinctus* were: 550 ms for exposure, 70 kV for voltage, 200 µA for current, and 18.00 µm for resolution (filter: Al 0.5 mm). The acquisition settings used for *Yurlunggur* were: 1770 ms for exposure, 85 kV for voltage, 118 µA for current, and 17.00 µm for resolution (filter: Al 0.5 mm). Three-dimensional visualization and imaging were carried out using Avizo Lite v. 9.0 (Thermo Scientific) and Dragonfly v. 4.1 (Object Research Systems Inc.). The left dentary of *Portugalophis lignites* [Museu Geológico, Lisboa, Portugal (MG-LNEG) 28094] was scanned using a Skyscan 1172 (Bruker) µCT scanner at the Instituto Superior Técnico, Portugal under 700 ms for exposure, 80 kV for voltage, 125 µA for current, and 4.164 µm as final resolution. Three-dimensional visualization and imaging were carried out using CTVox® (Bruker) and Dragonfly.

We have included movies of the reconstructed µCT slices through pairs of resorbing and non-resorbing teeth of *Boa constrictor*, *Malayopython reticulatus*, *Yurlunggur* sp., and *Portugalophis lignites* (Supplementary Movies 1–8).

### TRAP staining of *Pantherophis guttatus* specimen
Slides were de-waxed in xylene, dehydrated in an ethanol series (100%, 100%, 90%, 70%, 50%, 2 min each), and then placed in two distilled water washes for 2 min each. TRAP stain was mixed by dissolving N-N-Dimethylformamide in Naphthol-AS-TR-phosphate. A pH 5.2 Acetate buffer (glacial acetic acid, sodium acetate, and water) was added, followed by sodium tartrate and Fast Red TR salt. These samples were then counterstained with haematoxylin. These sections are accessioned at the Museum of Life Sciences (King's College London) under MoLS X282 and are publicly accessible.

### H&E staining of *Varanus* and *Boa* specimens
Slides were first de-waxed in two Toluene washes (5 min each), dehydrated in an ethanol series (100%, 100%, 90%, 70%, 50%, 2 min each), and then placed in distilled water for 2 min. Slides were then placed in Hematoxylin Gill III (Surgipath/Leica) for 2 min, followed by a 15-min rinse under cold tap water. After rinsing, the slides were placed in 70% ethanol (2 min), and then in Eosin (Surgipath/Leica) for 30 seconds. Slides were then placed in two 100% ethanol washes (2 min each), followed by two submersions in Toluene (2 min each). Coverslips were glued to the slides with DPX fixative.

### H&E staining of *Anilios bicolor* specimen
Slides were dried in an oven for 10 min, followed by two 3-min xylene washes, followed be two 1-min absolute ethanol washes, 1 min in 70% ethanol, and 1 min in tap water. Slides were then placed in Dako Harris Heamatoxylin for 1 min, followed by a 1-min tap water wash. Slides were then placed in acid alcohol for 30 s, then Dako Bluing Buffer for 1 min, followed by a 1-min tap water wash and 1 min in 70% ethanol. Eosin staining was done for 4.5 min in Dako Eosin Y Phyloxine, followed by two 1-min absolute ethanol and two 1-min xylene washes. Slides were then submerged in Histoclear and then coverslipped.

### Masson's trichrome staining of *Iguana* specimens
Slides were first de-waxed in two Toluene washes (5 min each), dehydrated in an ethanol series (100%, 100%, 90%, 70%, 50%, 2 min each), and then placed in distilled water for 2 min. Slides were then placed in Hematoxylin Gill III (Surgipath/Leica) for 1 min, followed by a 15-min rinse under cold tap water. After the rinse, the slides were placed in Ponceau-acid-fuchsin (2 min), followed by a series of distilled water rinses for 1 min. Slides were then placed in 1% phosphomolybdic acid (5 min) and then submerged in acetic aniline blue (3 min). The slides were submerged in another treatment of 1% phosphomolybdic acid (5 min), followed by 1% aqueous acetic acid (3 min), 95% ethanol (2 min), 100% ethanol (twice for 2 min), and two toluene washes (2 min each). Coverslips were applied using DPX fixative.

The staining protocols for the remaining slides can be found in Budney[56].

### Statistics and reproducibility
Sections presented in Figs. 1a–k, 2b–h, 3g–k, 4e, f, 5b–d, h–j, and Supplementary Figs. 1a–d, 2a–c, 4d–f, 8a–f, 9a–c were all taken from single individuals for each species and are taken to represent the normal condition for each taxon.

### Reporting summary
Further information on research design is available in the Nature Portfolio Reporting Summary linked to this article.

## Data availability
All of the thin section images generated in this study are in the main manuscript and the Supplementary Information. The µCT data for *Boa constrictor* and *Yurlunggur* used in this study are available on Morphosource (Project: Teeth of Serpentes https://www.morphosource.org/projects/000358042). The µCT data for *Malayopython reticulatus*, *Boiga dendrophila*, *Crotalus atrox*, *Anilios bicolor*, and *Portugalophis lignites* used in this study are available on Morphosource (Project: Modern and fossil snake tooth replacement https://www.morphosource.org/projects/000395615?locale=en). For specimen numbers and institutions, see Supplementary Data 1.

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

## Acknowledgements

We thank A. Oatway for assistance with thin sectioning and staining protocols at the University of Alberta. L. W. Kline donated all of the *I. iguana* samples used in this study. I. Paparella assisted with some of the sectioning of the *I. iguana* samples. We also thank O. Vernygora and Y. Wong for assistance with µCT scanning and M. Gingras for the use of the µCT scanner at the University of Alberta. Thanks to G. Sales, S. Ryder (MoLS), T. Ponce Leão, R. Dias, J. Moita, and J. Sequeira (MG-LNEG), and J. Streicher (BMNH) for collections access. We thank Adelaide Microscopy and Microscopy Australia for access to the µCT scanning equipment and Nanozoomer at the University of Adelaide (Adelaide, Australia); A. Labridinis and R. Williams for assistance; K. Batra, Y. Ciuk, A., and E. Schneider from Histology Services at the Faculty of Health and Medical Sciences, University of Adelaide. M. S. Y. Lee, M. N. Hutchinson, K. Sanders, and E. Sheratt provided logistical support for µCT scanning and histology of SAMA specimens. We also thank the LNEG and Museu Geológico as well as R. Dias, J. Moita, and J. Sequeira for their assistance in scanning the *Portugalophis* dentary. O. Zahradníček provided specimens of *C. atrox* and *B. dendrophila*. A.R.H.L. was supported by an NSERC Postdoctoral Fellowship and by a European Research Council MSCA-IF fellowship (no. 894331). A.P. was supported by the Australian Research Council (no. DP200102328) and a Visiting Fellowship from the Faculty of Science (University of Alberta). R.A. was supported by an FCT-AGA KAHN Development Network Grant (no. 333206718). A.S.T. was supported by a BBSRC grant on tooth replacement mechanisms (BB/W00240X/1). M.W.C. is supported by an NSERC Discovery Grant (no. 23458) and a Biological Sciences Chair's Research Allowance (University of Alberta).

## Author contributions

A.R.H.L. and A.P. conceived the project and collected most of the histological, osteological, and CT data. A.R.H.L. wrote the manuscript and prepared the figures with assistance from A.P. N.A. performed the thin sections and TRAP staining of the corn snake and provided CT data. R.A. and M.F.C.P. provided CT scans of *Portugalophis*. A.S.T. and M.W.C. provided lab materials, specimens, and assisted in interpreting the data. All authors edited previous drafts of the manuscript.

## Competing interests

The authors declare no competing interests.
