## [Peer Review File · Nature Communications]

A conserved tooth resorption mechanism in modern and fossil snakesReviewers' Comments:

Reviewer #1:

Remarks to the Author:

The manuscript of LeBlanc et al. provides novel and substantial information on tooth resorption in snakes. The paper is well written and the results supported by the figures and videos accompanying it. I must note that there is no full consensus that *Portugalophis* is definitely a snake, but I understand that this is of course beyond the scope of this paper. Nevertheless, the conclusions from this study will be of importance for the future phylogenetic assessment of this genus (and other "parviraptorids"). I have only some very minor remarks:

Line 110 (and elsewhere): the current combination for this species is *Aspidoscelis exsanguis*.

Line 127 (and elsewhere): the current combination for this species is *Malayopython reticulatus*.

Line 212: I suggest using the term "caenophidian" instead of "colubroid", as the latter has different concepts according to different authors. I see that you have explained it anyway in your Fig. 3, but it would be better still to avoid any misconceptions.

Line 590: The age of the oldest *Yurlunggur* were supposed to be Oligocene but recent radiometric dating has demonstrated it to be instead early Miocene (Woodhead et al. 2016). So please modify the sentence here as the genus is known from the Miocene-Pleistocene (unless your specimen truly originates from an older, well dated, locality – if so, then it is the oldest record of the genus and this must be mentioned somewhere anyway).

Woodhead, J., S. J. Hand, M. Archer, I. Graham, K. Sniderman, D. A. Arena, K. H. Black, H. Godthelp, P. Creaser and E. Price. 2016. Developing a radiometrically-dated chronologic sequence for Neogene biotic change in Australia, from the Riversleigh World Heritage Area of Queensland. *Gondwana Research*, 29:153–167.

Figure 3: I would suggest using *Constrictores* (sensu Georgalis and Smith 2020) instead of *Henophidia*, for two reasons: 1) in many recent phylogenies, *Henophidia* is used to encompass also *Caenophidia* (e.g., Gauthier et al. 2012; Figueroa et al. 2016); 2) *Boa* and *Malayopython* that you had in your sample are members of *Constrictores*.

Figueroa, A., McKelvy, A. D., Grismer, L. L., Bell, C. D., and Lailvaux, S.P. 2016. A Species-level phylogeny of extant snakes with description of a new colubrid subfamily and genus. *PLoS One*, 11:e0161070.

Gauthier, J. A., Kearney, M., Maisano, J. A., Rieppel, O., & Behike, A. D. B. (2012). Assembling the squamate tree of life: perspectives from the phenotype and the fossil record. *Bulletin of the Peabody Museum of Natural History*, 53, 3–308.

Georgalis, G.L. and K.T. Smith. 2020. *Constrictores* Oppel, 1811 – the available name for the taxonomic group uniting boas and pythons. *Vertebrate Zoology* 70:291–304.

So on Line 585, it would be better to add in "phylogeny modified from "also "Georgalis and Smith 2020"

I am sure the authors will find it easy to implement these few modifications suggested. I consider that this paper will be of importance for further histological and anatomical studies on squamates. As such, I am recommending it for publication following some minor revision.

I am at your disposal for any further query

Yours sincerely

Dr. Georgios Georgalis

Reviewer #2:

Remarks to the Author:

The research presented here centers on histological aspects of tooth replacement in extant snakes and in two fossil forms: the Australian "madtsoiid" snake *Yurlunggur* and the parviraptorid squamate *Portugalophis*. The authors compare their histological results to two representatives of the higher-level clades *Iguania* and *Anguimorpha*, which compose the clade *Toxicofera* along with snakes.

According to the authors, snakes are unique in lacking external signs of tooth resorption during tooth replacement (i.e., expanded resorption pits). Differently from all other squamates, dentine resorption of the functional tooth occurs only from within the pulp cavity through the unusual action of odontoclasts. Their results seem to reveal that internal tooth resorption is easily detectable via non-destructive Ct scanning, and that similar traces are present in Yurlunggur and Portugalophis.

The authors reveal an interesting mode of replacement that explains the lack of expanded external resorption in extant snake teeth. However, their conclusions are undermined by the very limited taxon sampling used, which leaves out of their study a number of key taxa. For instance, scolecophidians were not studied despite the fact that they are of crucial importance here since they retain fully pleurodont tooth attachment and upright replacement teeth, two plesiomorphic lizard conditions. Additionally, at least the anomalepidids among scolecophidians seem to retain large external resorption pits (Zaher and Rieppel, 1999: fig. 5). Similarly, the Cretaceous snake *Coniophis* cf. *precedens* seems to retain external signs of expanded resorption pits, visible in one of the few preserved teeth known (in maxilla AMNH 22127). On the 'lizard side,' as pointed out by the authors, the "lack of external resorption pits along the lingual surfaces of the teeth ... also occurs in helodermatids, *Lanthanotus*, *Varanus*, and some amphisbaenians" (page 3, lines 52-53). Other anguimorphs like *Shinisaurus* may also develop small resorption pits at the base of the teeth that fail to expand dorsally (Conrad, 2004). However, the authors preferred to present histological information only for the genus *Varanus*. There is no reasonable argument that can justify the exclusion from their analysis of these well-known extant lizards and of the whole scolecophidian radiation. Histological information on their replacement modes can lead to very different conclusions as to whether internal resorption is a synapomorphy of snakes (as they suggest) or, alternatively, a broader (Platynotan) or a more restricted (Alethinophidian) condition within Toxicofera.

As for the Jurassic parviraptorid *Portugalophis*, the authors seem to assume that there is a consensus on its snake affinities. However, its phylogenetic affinities are far from resolved, mainly as a result of the very fragmentary nature of this taxon. Parviraptor *estesi* was alternatively considered to be a stem Gekkonomorph (Conrad and Norell, 2006; Conrad, 2008) or the sister group to all squamates (Pyron, 2017). The authors should at least acknowledge the conflicting literature on this poorly-known group. Comparisons between *Portugalophis* and extant gekkonomorphs with similar conical and recurved teeth (e.g., *Lialis* and *Aprasia*) should be provided as a way to cover alternative hypotheses of relationships.

More specific comments are provided below:

On page 4, lines 66-71, the authors wrote "We then compare these data with histological thin sections and high-resolution computed tomography (CT) scans of several other snake species, and to thin sections of *Varanus* sp., *Iguana iguana*, and other lizards to identify key differences between the snake tooth replacement cycle and the two other canonical replacement modes in squamates (iguanaid- and varanid-type)." According to Supplementary Table 1, apart from *Varanus* sp. and *Iguana iguana*, only *Cnemidophorus exsanguis* and *Sauromalus ater* are added to the studied list. This hardly accounts for "several other snakes and lizards." A more reasonable mention could be "...two other lizards..."

On page 4, lines 73-79, the authors inform "Finally, we test this non-destructive method on two key fossil taxa: the archaic madtsoiid snake *Yurlunggur* from the early Miocene of Australia and one of the oldest snake fossils, *Portugalophis* lignites from the Jurassic of Portugal. These data reveal that snakes possess an unusual form of tooth replacement that differs from current models of physiological tooth resorption in reptiles, and is even present in the oldest snake lineage, which predates the evolutionary loss of limbs." Alternative hypotheses of relationships of *Portugalophis* should be mentioned here. Additionally, the authors provided as Supplementary information only Micro-Ct scan vertical slices of the teeth of *Yurlunggur* and *Portugalophis*, but all the images providing the relevant information on odontoclast effects along the internal dentine surface are illustrated by the authors through transverse

slices. The authors should also provide the micro CT-scan movies depicting the transverse slices used in the manuscript. This will allow a more accurate reproducibility of their results.

Page 14, lines 288-307. The authors describe odontoclast action on the shedding of functional teeth in *Varanus* that may likely be considered similar to what was described in snakes. Odontoclast action at the base of the tooth, along the region with plicidentine, may likely represent a slightly different but homologous condition to snakes. This reinforces the need of expanding their comparison to the other extant platynotans mentioned above.

Page 17, lines 374-377. The authors reach several conclusions regarding *Portugalophis* that depend directly on their very limited taxon sampling. Whether *Portugalophis* is a snake is still debatable and the authors should avoid definitive conclusions without providing broader comparisons with other groups of squamates (e.g., Gekkonomorphs, Dibamus, other anguimorphs) that can support more convincingly their statements.

In conclusion, the authors reach some very interesting results by studying the histological aspects of the process of tooth replacement in a limited number of extant alethinophidian snakes. However, whether these conclusions can be expanded to extant scolecophidians and to extinct snakes as a whole is a different issue. In my opinion, the authors failed to convincingly justify the generalizations advanced in their study, mainly because their taxon sampling is severely reduced.

Reviewer #3:

Remarks to the Author:

The authors have studied the tooth replacement cycle in a wide selection of snakes and other squamates. As expected from the highly regarded team of scientists that authored this study, the work is technically of very high quality. The techniques used include histology, histochemistry and microCT. The pattern of resorption is internal, both in ancestral and modern snakes. While I like the quality of this descriptive study, I do think it has rather a limited and specialised audience. And although it identifies odontoclast activity as being crucial to the process, I question whether this is really enough of a paradigm shift at this time.

Reviewer #1 (Remarks to the Author):

The manuscript of LeBlanc et al. provides novel and substantial information on tooth resorption in snakes. The paper is well written and the results supported by the figures and videos accompanying it. I must note that there is no full consensus that *Portugalophis* is definitely a snake, but I understand that this is of course beyond the scope of this paper.

We address this comment below in the reply to R2's criticism.

Nevertheless, the conclusions from this study will be of importance for the future phylogenetic assessment of this genus (and other "parviraptorids").

I have only some very minor remarks:

Line 110 (and elsewhere): the current combination for this species is *Aspidoscelis exsanguis*.

RESPONSE: We have corrected this.

Line 127 (and elsewhere): the current combination for this species is *Malayopython reticulatus*.

RESPONSE: We have corrected this.

Line 212: I suggest using the term "caenophidian" instead of "colubroid", as the latter has different concepts according to different authors. I see that you have explained it anyway in your Fig. 3, but it would be better still to avoid any misconceptions.

RESPONSE: We have corrected this.

Line 590: The age of the oldest *Yurlunggur* were supposed to be Oligocene but recent radiometric dating has demonstrated it to be instead early Miocene (Woodhead et al. 2016). So please modify the sentence here as the genus is known from the Miocene-Pleistocene (unless your specimen truly originates from an older, well dated, locality – if so, then it is the oldest record of the genus and this must be mentioned somewhere anyway).

Woodhead, J., S. J. Hand, M. Archer, I. Graham, K. Sniderman, D. A. Arena, K. H. Black, H. Godthelp, P. Creaser and E. Price. 2016. Developing a radiometrically-dated chronologic sequence for Neogene biotic change in Australia, from the Riversleigh World Heritage Area of Queensland. *Gondwana Research*, 29:153–167.

RESPONSE: We have corrected this and provide the citation in the figure caption.

Figure 3: I would suggest using *Constrictores* (sensu Georgalis and Smith 2020) instead of *Henophidia*, for two reasons: 1) in many recent phylogenies, *Henophidia* is used to encompass also *Caenophidia* (e.g., Gauthier et al. 2012; Figueroa et al. 2016); 2) *Boa* and *Malayopython* that you had in your sample are members of *Constrictores*.

Figueroa, A., McKelvy, A. D., Grismer, L. L., Bell, C. D., and Lailvaux, S.P. 2016. A Species-level phylogeny of extant snakes with description of a new colubrid subfamily and genus. *PLoS One*, 11:e0161070.

Gauthier, J. A., Kearney, M., Maisano, J. A., Rieppel, O., & Behike, A. D. B. (2012).

Assembling the squamate tree of life: perspectives from the phenotype and the fossil record. *Bulletin of the Peabody Museum of Natural History*, 53, 3–308.

Georgalis, G.L. and K.T. Smith. 2020. *Constrictores* Oprel, 1811 – the available name for the

taxonomic group uniting boas and pythons. *Vertebrate Zoology* 70:291–304.

So on Line 585, it would be better to add in “phylogeny modified from” also “Georgalis and Smith 2020”

RESPONSE: We have corrected this in Figure 3 and provide the additional citation in the figure caption.

I am sure the authors will find it easy to implement these few modifications suggested. I consider that this paper will be of importance for further histological and anatomical studies on squamates. As such, I am recommending it for publication following some minor revision. I am at your disposal for any further query

Yours sincerely

Dr. Georgios Georgalis

Reviewer #2 (Remarks to the Author):

The research presented here centers on histological aspects of tooth replacement in extant snakes and in two fossil forms: the Australian “madtsoiid” snake *Yurlungur* and the parviroptorid squamate *Portugalophis*. The authors compare their histological results to two representatives of the higher-level clades Iguania and Anguimorpha, which compose the clade Toxicofera along with snakes.

According to the authors, snakes are unique in lacking external signs of tooth resorption during tooth replacement (i.e., expanded resorption pits). Differently from all other squamates, dentine resorption of the functional tooth occurs only from within the pulp cavity through the unusual action of odontoclasts. Their results seem to reveal that internal tooth resorption is easily detectable via non-destructive Ct scanning, and that similar traces are present in *Yurlungur* and *Portugalophis*.

The authors reveal an interesting mode of replacement that explains the lack of expanded external resorption in extant snake teeth.

However, their conclusions are undermined by the very limited taxon sampling used, which leaves out of their study a number of key taxa. For instance, scolecophidians were not studied despite the fact that they are of crucial importance here since they retain fully pleurodont tooth attachment and upright replacement teeth, two plesiomorphic lizard conditions. Additionally, at least the anomalepidids among scolecophidians seem to retain large external resorption pits (Zaher and Rieppel, 1999: fig. 5).

RESPONSE: The reviewer is misrepresenting Zaher and Rieppel (1999). We quote the final sentence of the discussion in the aforementioned paper before making our point:

“Snakes never develop resorption pits.” (Zaher and Rieppel, 1999:12)

The reviewer mistook nutritive/alveolar foramina (identified by Zaher and Rieppel) at the base of the teeth of the anomalepidid *Liotyphlops* for resorption pits.

The Reviewer also wrote that we have “very limited” taxonomic sampling and that they would suggest including additional “*key taxa*”- with particular reference to scolecophidian snakes. To support the importance of adding scolecophidians to our comparison the reviewer argues that these snakes still retain primitive lizard features, and cites the figure from Zaher and Rieppel (1999). However, as discussed above these snakes do not have resorption pits or other signs of external resorption, so their tooth replacement must be the same as in other snakes, which is the focal point of this study.

Our taxonomic sampling currently consists of a broad range of alethinophidian snakes (elapids, colubrids, acrochordids, and the constrictors *Boa* and *Malayopython*). Based on over 60 years of investigations, all snakes replace their teeth in the same way. This was never in doubt (see Edmund 1960, Zaher and Rieppel, 1999). Our paper is the first to show how they do it.

Additionally, including scolecophidians in our sample would be extremely difficult for two reasons:

- (A) The extremely small size of scolecophidians makes observations of microscopic details of their teeth (<0.5 mm tall crowns) very challenging by conventional non-destructive means and would require further (destructive) sample preparation to allow for more precise, high-resolution tomography (for example, at a tomography beamline at a synchrotron facility).

Conventional lab-based microCT would seem like an alternative non-destructive option, but the resolution is not high enough to visualize microscopic features (Howship’s lacunae) in such small teeth. We examined several openly available scans (e.g., Morphosource) as well as scans made by one of the coauthors of this manuscript (down to a resolution of 3 microns; AP pers. Obs.), and it is impossible to achieve enough resolution to visualize internal resorption (Howship’s lacunae) in these teeth, because they are predicted to be less than 3 microns in diameter.

- (B) Studying scolecophidian tooth replacement more precisely requires histological sampling and would therefore require destructive sampling of museum specimens from rare or endangered species. For example, Anomalepidids such as *Liotyphlops* are extremely rare in museum collections and destructive sampling (histology) is therefore not allowed. Further complicating this is the small number of teeth in scolecophidians, meaning that the odds of catching a tooth at the right stage from a single section is far more unlikely than in other snakes. We would therefore require destructive sampling of multiple individuals to be sure we capture the same level of detail as we present here for our snake sample.

However, please bear in mind that resorption pits and signs of external resorption are still absent in all scolecophidians, as already established in previous studies (e.g., Zaher and Rieppel, 1999). Moreover, our study of two key snake fossils provide valuable insight here. Internal resorption in the fossil snake *Yurlunggur* and, more importantly, the Jurassic fossil *Portugalophis*, would suggest that the appearance of internal tooth resorption mechanisms in snakes predates the split between scolecophidians and alethinophidians.

We have added a statement about this in the discussion section of the revised manuscript:

Even scolecophidian snakes, despite their pleurodont type of tooth implantation¹⁶, lack resorption pits^{16,50} and may thus follow a resorption mechanism similar to what we have described here based on our broad sample of fossil and extant snakes. However, confirming the presence of internal tooth resorption in extant scolecophidian species will require histological sampling rather than conventional μ CT approaches, because of their extremely small body sizes and small teeth (often less than 500 μ m tall; AP pers. obs.).

Similarly, the Cretaceous snake *Coniophis cf. precedens* seems to retain external signs of expanded resorption pits, visible in one of the few preserved teeth known (in maxilla AMNH 22127).

RESPONSE: We are not aware of any jaw materials assigned to *Coniophis cf. precedens*. The only cranial material assigned to *Coniophis precedens* has been in the recent work of Longrich et al. (2012) and AMNH 22127 is not a maxillary element listed in that paper (three maxillae were referred to *C. precedens* by Longrich et al., including AMNH 22413, and UCMP 49999 and 53935).

Perhaps the Reviewer meant to refer to AMNH 22413? In any case, in the absence of a photograph of 22127, or reference, we cannot address the “pit”, though it seems likely the specimen is that of a non-snake squamate, similar to several of the specimens assigned to *Coniophis* by Longrich et al. (2012) as argued in Caldwell et al. (2015, suppl.). This would make for an important future investigation.

On the ‘lizard side,’ as pointed out by the authors, the “lack of external resorption pits along the lingual surfaces of the teeth ... also occurs in helodermatids, Lanthanotus, Varanus, and some amphisbaenians” (page 3, lines 52-53). Other anguimorphs like *Shinisaurus* may also develop small resorption pits at the base of the teeth that fail to expand dorsally (Conrad, 2004).

RESPONSE: Small resorption pits in *Shinisaurus* are still resorption pits. These are the consequences of external resorption. The pits enlarge as the replacement tooth enlarges and encroaches upon the base of the older, functional tooth. This is how typical tooth replacement (involving resorption pits) progresses. This taxon cannot be considered as having exclusively internal tooth resorption as snakes do, because it clearly develops these resorption pits in association with replacement teeth, as shown in the image below.

Shinisaurus crocodilurus (SAMA R66666), red arrows points at large resorption pit.

In addition, the legless anguid *Anguis fragilis* also has clear resorption pits, which we now cite in the main text (Cooper, 1966).

However, the authors preferred to present histological information only for the genus *Varanus*. There is no reasonable argument that can justify the exclusion from their analysis of these well-known extant lizards and of the whole scolecophidian radiation.

RESPONSE: We do not need to include more lizards with varanid-type tooth replacement, because it is a well-established phenomenon across varanoid lizards, see for example Kearney and Rieppel (2006:344):

“Varanoids, however, do not develop resorption pits. Instead, older teeth are usually shed due to erosion of the tooth bases by osteoclastic activity.”

Varanus is simply one of the more accessible genera to use in our comparison with snakes, but the same phenomenon occurs in the other varanoids, *Heloderma* and *Lanthanotus*. See also Edmund’s (1960) seminal monograph on tooth replacement (which we cite in the main text):

“*Heloderma* exhibits the typical varanid method of replacement. The old tooth remains intact until the replacement tooth is about half grown, then is shed, apparently by detachment of its relatively intact base from the jaw. No trace of formation of basal pits was seen in any specimen studied (our emphasis)”. (Page 81)

“The replacement process in *Lanthanotus* is practically identical with that seen in *Heloderma* and is clearly of the varanid pattern. As in *Heloderma* the old tooth is shed without development of a basal pit [our emphasis], probably by resorption of the ankylosing bone”. (Page 85)

Histological information on their replacement modes can lead to very different conclusions as to whether internal resorption is a synapomorphy of snakes (as they suggest) or, alternatively, a broader (Platynotan) or a more restricted (Alethinophidian) condition within Toxicofera.

RESPONSE: The reason behind the exclusion of scolecophidians has already been discussed above. Tooth replacement in the other taxa has already been extensively studied (e.g. Edmund, 1960; Zaher and Rieppel, 1999; Kearney and Rieppel, 2006).

Additionally, *Shinisaurus* and *Lanthanothus* are rare, endangered taxa that are not easy to access. There are very few specimens in museum collections, and destructive sampling (histology) of fluid-preserved specimens is unfortunately out of the question in most cases. While acknowledging the fact that our taxon sampling could be extended in an ideal world without endangered species, we firmly believe that it is sufficient to support our general conclusions. *Varanus* is the consensus model for “varanid-type” tooth replacement (Edmund, 1960; Zaher and Rieppel, 1999; Kearney and Rieppel, 2006).

Moreover, there is no consensus regarding the closest sister group to snakes among lizards, but molecular studies tend to agree that it is probably the clade inclusive of both Iguania and Anguimorpha (e.g. Pyron et al. 2013; Zheng and Wiens, 2016; Harrington and Reeder, 2017; Burbrink et al. 2020), where these clades are grouped with snakes in the Toxicofera. None of these studies support the old notion of a clade named “Platynota”, mentioned by the Reviewer (“Platynota” is currently considered a polyphyletic assemblage inclusive of anguimorphans, mosasaurians, and snakes). Currently, apart from Iguania+Anguimorpha, the only alternative sister group to snakes would be Mosasauria, as found for example in Reeder et al. (2015). However, because mosasaurs are known to all possess resorption pits (like most non-snake lizards), then following this evolutionary hypothesis internal resorption would be clearly a synapomorphy of snakes.

We wanted to test the more complex Toxicoferan scenario and thus sampled representatives from Iguania and Anguimorpha. These were shown to lack the internal tooth resorption typical of snakes. Because of the phylogenetic distance between the sampled taxa, it is safe to assume that also other taxa within these clades lack exclusively internal tooth resorption as well. On the other hand, all of the snakes we sampled show only internal tooth resorption.

While we did not sample scolecophidians, we were explicit about it. Regardless of that, our sample is still broad enough to state that internal tooth resorption is a shared feature of at least Alethinophidian snakes, but possibly of all snakes based on its presence in the basal fossil snake *Yurlunggur* and in the dentary of *Portugalophis*.

As for the Jurassic parviraptorid *Portugalophis*, the authors seem to assume that there is a consensus on its snake affinities. However, its phylogenetic affinities are far from resolved, mainly as a result of the very fragmentary nature of this taxon. *Parviraptor estesi* was alternatively considered to be a stem Gekkonomorph (Conrad and Norell, 2006; Conrad, 2008) or the sister group to all squamates (Pyron, 2017). The authors should at least acknowledge the conflicting literature on this poorly-known group.

RESPONSE: The reviewer is conflating *Portugalophis* with *Parviraptor*. There is no conflicting literature on the snake affinities of *Portugalophis*. The reviewer cites a work from 2012, but that reference refers to *Parviraptor*, not *Portugalophis*. *Portugalophis*

lignites was described and named in 2015 (Caldwell et al. 2015), and since then (7 years) the snake affinity of *Portugalophis* has never been rebutted by published criticisms followed by replacement phylogenetic hypotheses. On the contrary, the snake affinity of *Portugalophis* was supported in a recent study by Harrington and Reeder (2017), which we now cite in the main text.

We also already acknowledge, with supporting citations, the debates surrounding the identity and evolutionary history of early snakes: “*Despite their unique bauplan, the evolutionary relationships of snakes to other lizards have been extensively debated²⁻⁹. Furthermore, the fossil record of early snakes is fragmentary, represented by isolated vertebrae and partial jaws, which hinders our ability to confidently identify fossil snakes or establish the relative timing of the appearance of key snake features^{7,9,10}*” (

Regarding *Parviraptor* and *Portugalophis*, while both taxa were considered Jurassic snakes in Caldwell et al. (2015) there is no evidence that they are closely related. We were able to sample the dentary of *Portugalophis lignites* from Portugal and found evidence for internal resorption similar to that in extant snakes. We have made no such statement regarding *Parviraptor*, but we hope that this can be investigated in this fossil in the future.

Comparisons between *Portugalophis* and extant gekkonomorphs with similar conical and recurved teeth (e.g., *Lialis* and *Aprasia*) should be provided as a way to cover alternative hypotheses of relationships.

RESPONSE: We would like to reiterate that so far nobody in the literature has argued that *Portugalophis* could be a gekkotan. Additionally, we do not see the similarity between *Portugalophis*, a snake with long, sharp, very recurved teeth, and gekkotans such as *Lialis* or *Aprasia* (see images of their dentaries below). We believe that discussing these obvious dissimilarities is unnecessary and beyond the scope of our paper.

We assume that the reviewers’ comment is based on Conrad’s (2008) recovery of *Parviraptor*, (which is not *Portugalophis*) as a gekkotan (as mentioned earlier). However, there is no report of any author interpreting *Portugalophis* as a gekkotan, and this is the only Jurassic fossil discussed in our manuscript.

Per R2's request, here are comparisons between *Portugalophis*, the snake *Cylindrophis*, and two gekkotans mentioned by the reviewer. External resorption pits are obvious in the gekkotan taxa (black arrows). The pits are missing in snakes, and as we show in our study they are replaced by internal resorption. Please note that resorption pits are large excavations of various sizes (depending on the stage of resorption) at the base of a tooth and typically affect the crown. The consistently small foramina at the base of snake teeth, which do not affect the crown, are nutritive foramina. Reviewer 2 misidentified the latter for the former when looking at images of scolecochidians teeth.

Note the presence of resorption pits at the bases of the teeth in the two gekkotans *Lialis* and *Aprasia*. We do not show this in the main manuscript, because it is common knowledge in the literature that gekkotans produce resorption pits (e.g., Edmund, 1960), whereas snakes do not. This would also detract from the central message of the paper.

More specific comments are provided below:

On page 4, lines 66-71, the authors wrote “We then compare these data with histological thin sections and high-resolution computed tomography (CT) scans of several other snake species, and to thin sections of *Varanus* sp., *Iguana iguana*, and other lizards to identify key differences between the snake tooth replacement cycle and the two other canonical replacement modes in squamates (iguanaid- and varanid-type).” According to Supplementary Table 1, apart from *Varanus* sp. and *Iguana iguana*, only *Cnemidophorus exsanguis* and *Sauromalus ater* are added to the studied list. This hardly accounts for “several other snakes and lizards.” A more reasonable mention could be “...two other lizards...”

RESPONSE: We have added histological data and macrophotography of the early stages of the replacement cycle in several more lizard species, now listed in an updated Supplementary Table 1, and Suppl. Fig. 6. We also added additional citations for anguoid tooth replacement (Cooper, 1966) to the main text.

However, we reiterate that iguanid- and varanid-type tooth replacement are extremely well-documented and we do not see the necessity of providing further extensive

coverage of what is already known. The strength of this comparative study is in the detail in which we document the cellular and tissue-level processes involved in tooth replacement between snakes and other lizards. As expected, these additional squamate taxa we have added all replace their teeth in the more typical iguanid fashion.

On page 4, lines 73-79, the authors inform “Finally, we test this non-destructive method on two key fossil taxa: the archaic madtsoiid snake *Yurlunggur* from the early Miocene of Australia and one of the oldest snake fossils, *Portugalophis lignites* from the Jurassic of Portugal. These data reveal that snakes possess an unusual form of tooth replacement that differs from current models of physiological tooth resorption in reptiles, and is even present in the oldest snake lineage, which predates the evolutionary loss of limbs.” Alternative hypotheses of relationships of *Portugalophis* should be mentioned here.

RESPONSE: Again, there are no published alternative hypotheses for the placement of *Portugalophis* that we are aware of.

Additionally, the authors provided as Supplementary information only Micro-Ct scan vertical slices of the teeth of *Yurlunggur* and *Portugalophis*, but all the images providing the relevant information on odontoclast effects along the internal dentine surface are illustrated by the authors through transverse slices.

RESPONSE: We use Supplementary Fig. 5 only to enlarge the images of the internal scalloping in the *Portugalophis* scans to show it more clearly, but it is already fully labelled in the main text Fig. 3 in two planes. We have added an additional video showing a series of cutaways of this *Portugalophis* tooth to show the smooth outer surface of the tooth (lack of tooth resorption) and the rough internal walls (we argue being caused by internal resorption). The resorption structures in *Portugalophis* are already labelled in vertical slices in the main text Figure 3B.

The authors should also provide the micro CT-scan movies depicting the transverse slices used in the manuscript. This will allow a more accurate reproducibility of their results.

RESPONSE: We have added a large CT scan movie for *Portugalophis* as well. Please also note that for reproducibility of our results, readers will be able to download the actual CT files from Morphosource and examine the scans for themselves in all directions.

Page 14, lines 288-307. The authors describe odontoclast action on the shedding of functional teeth in *Varanus* that may likely be considered similar to what was described in snakes. Odontoclast action at the base of the tooth, along the region with plicidentine, may likely represent a slightly different but homologous condition to snakes. This reinforces the need of expanding their comparison to the other extant platynotans mentioned above.

RESPONSE: We have clarified varanid-type tooth replacement with the addition of new images of other *Varanus* teeth from additional specimens in Fig. 4 and an entirely new Supplementary Fig. 9 to reinforce the main conclusion of this paper: tooth resorption begins on the outer surfaces of the teeth in these lizards. It begins internally in snakes. There is no similarity to the snake tooth replacement cycle here. Rieppel (1978) and Zaher and Rieppel (1999) also came to the same conclusion regarding the difference between a “varanid”-type tooth replacement mode and those of snakes, which (inexplicably at the time) lack resorption pits. We have rectified this by determining how snakes do it and demonstrate that it is identifiable under the right circumstances via non-destructive CT scanning.

Finally, we would like to point out to the reviewer that snakes also have plicidentine (Palci et al, 2021), and yet they do not have the same replacement mode as varanids.

Page 17, lines 374-377. The authors reach several conclusions regarding *Portugalophis* that depend directly on their very limited taxon sampling. Whether *Portugalophis* is a snake is still debatable and the authors should avoid definitive conclusions without providing broader comparisons with other groups of squamates (e.g., *Gekkonomorphs*, *Dibamus*, other *anguimorphs*) that can support more convincingly their statements.

RESPONSE: We are unsure of the reviewer’s familiarity with the literature on this topic, but this is a mistaken belief that the replacement modes in gekkonomorphs and anguimorphs are unknown. From Edmund (1960) and Cooper (1966) onwards, there are detailed, illustrated, and photographed accounts of how lizards in these groups replace their teeth. Gekkonomorphs and anguimorphs replace their teeth very differently from snakes and we summarize the relevant literature on this in the main text. We cite key references whenever we can regarding this form of tooth replacement in the main text.

Regarding the snake affinities of *Portugalophis* and the reviewer’s perceived similarity of *Portugalophis* and gekkotans please see our previous reply.

Reviewer #3 (Remarks to the Author):

The authors have studied the tooth replacement cycle in a wide selection of snakes and other squamates. As expected from the highly regarded team of scientists that authored this study, the work is technically of very high quality. The techniques used include histology, histochemistry and microCT. The pattern of resorption is internal, both in ancestral and modern snakes. While I like the quality of this descriptive study, I do think it has rather a limited and specialised audience. And although it identifies odontoclast activity as being crucial to the process, I question whether this is really enough of a paradigm shift at this time.

We appreciate the compliments offered by this reviewer to our team and our work, but are puzzled by the lack of enthusiasm for our findings. Here we are presenting evidence of a unique mode of tooth replacement that characterizes one of the most iconic groups of organisms, snakes, which owe their popularity largely to their sharp teeth and fangs.

Snakes can replace their teeth and fangs extremely rapidly, but until now the mechanism behind this fast replacement was completely unknown. The repercussions of our findings are multiple:

(1) snakes *uniquely* replace their teeth via an internal resorption mechanism that weakens old teeth from within, without ever forming a resorption pit. To the best of our knowledge this has not been reported in any other vertebrates, and is by itself quite an exciting discovery.

(2) internal tooth resorption leaves telltale scalloping along the inner walls of functional teeth in snakes, which is detectable by non-destructive μ CT scans of isolated jaws. This is extremely useful in helping with the identification of isolated and fragmentary snake remains in the fossil record. Identification of these fragmentary remains, as reviewer 2 pointed out, can be quite controversial in the absence of easily identifiable and unique snake features.

(3) we detected this unique snake-type tooth replacement in the lower jaw of one of the oldest snakes known (*Portugalophis*, from the Upper Jurassic of Portugal), which indicates that such feature was one of the first hallmarks of snakes, possibly even before limb loss.

Furthermore, questions of snake origins, descriptions of key snake fossils, and interpretations of snake evolutionary relationships are abundant in non-specialist journals, because they are popular topics (we have cited many of them in the present manuscript):

Caldwell, M. W. & Lee, M. S. Y. A snake with legs from the marine Cretaceous of the Middle East. *Nature* 386, 705–709 (1997).

Lee, M. S. Y. The phylogeny of varanoid lizards and the affinities of snakes. *Philos. Trans. R. Soc. B Biol. Sci.* 352, 53–91 (1997).

Tchernov, E., Rieppel, O., Zaher, H., Polcyn, M. J. & Jacobs, L. L. A fossil snake with limbs. *Science* 287, 2010–2012 (2000).

Caldwell, M. W., Nydam, R. L., Palci, A. & Apesteguía, S. The oldest known snakes from the Middle Jurassic-Lower Cretaceous provide insights on snake evolution. *Nat. Commun.* 6, 5996 (2015).

Martill, D. M., Tischlinger, H. & Longrich, N. R. A four-legged snake from the Early Cretaceous of Gondwana. *Science* 349, 416–419 (2015).

Garberoglio, F. F. *et al.* New skulls and skeletons of the Cretaceous legged snake Najash, and the evolution of the modern snake body plan. *Sci. Adv.* 5, eaax5833 (2019).

Longrich, N. R., Bhullar, B.-A. S. & Gauthier, J. A transitional snake from the Late Cretaceous period of North America. *Nature* 488, 205–208 (2012).

Palci, A. *et al.* Plicidentine and the repeated origins of snake venom fangs. *Proc. R. Soc. B Biol. Sci.* 288, 20211391 (2021).

Scanlon, J. D. Skull of the large non-macrostromatan snake *Yurlunggur* from the Australian Oligo-Miocene. *Nature* 439, 839–842 (2006).

That being said, we have taken this reviewer's criticism into consideration and hope that our modifications to the main text, particularly in the discussion and conclusions more clearly reflect the implications of our discovery. We have also added our own speculation for why snakes have evolved this peculiar form of tooth replacement, based on very recent literature on snake tooth replacement rates and our observation of the extensive tooth loss some snakes incur during feeding (see our new Suppl. Fig. 10).

In conclusion, we hope that we managed to convey the importance and novelty of our finding, that we convinced both reviewers and editor of its non-specialist nature, and that our paper would be a great fit to the broad audience of *Nature Communications*.

Sincerely,

A. R. H. LeBlanc and coauthors

Reviewers' Comments:

Reviewer #1:

Remarks to the Author:

Dear Editor, dear Authors,

I see that the comments I had raised during the first review round have now been addressed.

As a final remark, I would suggest mentioning, somewhere in the last paragraph (lines 411-420) or elsewhere, the recent publication of Zaher et al. (2022) who deals with macrophagy in early snakes, as it appears much relevant to your text about macrophagy.

Zaher, H., D.M. Mohabey, F.G. Grazziotin and J.A.W. Mantilla. 2022. The skull of *Sanajeh indicus*, a Cretaceous snake with an upper temporal bar, and the origin of ophidian wide-gaped feeding. *Zoological Journal of the Linnean Society*

In addition, in the sentence about plicidentine in *Varanus* "(the extensive plicidentine at the base of the teeth in *Varanus* partitions the pulp into a series of smaller chambers)" (lines 310-311), I would recommend mentioning also (in the same parenthesis) how is this respective plicidentine morphology in the extinct Eocene palaeowaranid lizard *Palaeovaranus*, which is a distant relative of *Varanus*. See CT-scans of *Palaeovaranus* teeth in Georgalis and Scheyer (2019:fig. 3).

Georgalis, G.L. and T.M. Scheyer. 2019. A new species of *Palaeopython* (Serpentes) and other extinct squamates from the Eocene of Dielsdorf (Zurich, Switzerland). *Swiss Journal of Geosciences* 112:383-417.

Also, the last sentence of the text, "Early snakes were likely already macrophagous in the Jurassic, and this dietary specialization may have evolved before the loss of limbs, which typifies the snake bauplan." seems somehow speculative, particularly when concerning the current non-consensus surrounding the true-snake affinities of the Jurassic forms (particularly *Parviraptor*). Therefore, perhaps some rephrase (or even deleting) of this sentence would be more appropriate.

Finally, I just point out that there are a few missing spaces appearing in the text (e.g., lines 113, 135) and misspelling of *Malayopython* in line 210 and *Dinilysia* in line 534, and some non-italicization of genera and species names in the references section.

As such, upon dealing with these ultimate few issues, I would recommend this paper for publication.

Reviewer #2:

Remarks to the Author:

I read carefully the authors rebuttal and thanks their effort in trying to clarify the main issues raised by the referees. Some of my questions were clarified appropriately and I am confident with the answers given. However, I am still not convinced by some of the main arguments put forward in this paper. I am especially puzzled by the resistance from the authors in trying to fill the obvious informational gaps in their work.

Therefore, some of my main criticisms remain unsolved. I will enumerate them below:

1) On scolecophidians – The unique resorption mechanism described by the authors represents the main and really important finding of this study. However, the authors failed to convincingly provide evidence that this mechanism is present in all snakes. Without further evidence from scolecophidians, there is no reason to view the novel mechanism as a synapomorphy of snakes, since it can well be an alethinophidian specialization instead. Filling the informational gap on scolecophidians is therefore crucial to their work. There is actually no published evidence [including Zaher and Rieppel (1999) mentioned by the authors] that addresses specifically the presence/absence of resorption pits in scolecophidians. Therefore, the authors should try to fill that gap of information since it is exactly what their study are meant to clarify. Quoting a sentence from Zaher and Rieppel (1999) simply does not solve the issue (this authors did not provide compelling evidence in that sense). What is seen in *Liotyphlops* can well be a resorption pit instead of an anomalously large nutritive foramen. The authors further consider that the "extremely small size of scolecophidians" and their rarity in collections prevent the study of tooth morphology on this group. I strongly disagree: scolecophidians

are well represented in many major collections, with sometimes huge series of individuals, including *Liotyplops* species. A significant number of *Typhlops* species grow to large size, and can be accessed for destructive studies if requested appropriately. The fact that the authors insist in avoiding to expand their study to scolecophidian tooth morphology weakens their arguments and conclusions.

2) On the fossils *Portugalophis* and *Yurlunggur* – The authors seem to consider that both fossils are key taxa to their study. However, recent studies have shown that *Yurlunggur* is likely nested within crown *Serpentes* and does not represent an early diverging snake (e.g., Zaher and Smith, 2020; Zaher et al. 2022). Therefore, it is not surprising to find in *Yurlunggur* a similar internal resorption mechanism than the one described by the authors in crown *alethinophidians*. As for *Portugalophis*, I am puzzled by the suggestion from the authors that *Portugalophis* and *Parviraptor* could be in some way distinct lineages, and that any comparison between these two taxa would be misleading. To my knowledge, no one has considered *Portugalophis* and *Parviraptor* in distinct position in the squamate tree either. In that sense, questions raised in the literature regarding the affinities of “*parviraptorids*” need to be addressed convincingly by the authors. In any case, the presence of internal resorption mechanism is not clearly visible in *Portugalophis* in my opinion. The ct slices and images provided by the authors are not convincing and the ct video does not correspond to the fifth tooth described in the paper but rather to two more posterior dentary teeth. Please, provide compelling evidence through videos and images of the tooth studied and of additional teeth in the dentary.

In summary, I think the study still suffers from serious flaws and fails to provide all the necessary evidence to support its claims. Additionally, the authors use two questionable fossils in support of their claims (with *Yurlunggur* being demonstrated to be far from the basal divergence of snakes and *Portugalophis* being a poorly preserved specimen with uncertain affinities with snakes). With that in mind, the overall impression is that their taxon sampling seems to be biased to their preferred hypothesis and does not test the broader picture. Finally, I would tend to agree with referee #3 that, even with the inclusion of scolecophidians to their sample, the paper is still written to reach a rather limited and specialized audience.

Additional References:

Zaher H, Smith KT. 2020. Pythons in the Eocene of Europe reveal a much older divergence of the group in sympatry with boas. *Biology Letters* 16: 20200735.

Zaher H, Mohabey DM, Graziotin FG, Mantilla JAW 2022. The skull of *Sanajeh indicus*, a Cretaceous snake with an upper temporal bar, and the origin of ophidian wide-gaped feeding, *Zoological Journal of the Linnean Society*, 2022;, zlac001, <https://doi.org/10.1093/zoolinnean/zlac001>

Reviewer #3:

Remarks to the Author:

The authors have, in my view, successfully defended the potential interest of their paper to the readership of *Nature Comms* by stressing that the internal pattern of resorption is unique to the clade studied. I find their arguments persuasive. They have also addressed very skilfully the comments of the other two reviewers. I therefore can now recommend publication.

Reviewer #4:

Remarks to the Author:

The manuscript gives a careful study of the manner in which snakes lose their teeth and draws comparisons to lizards. Particularly the histological and histochemical aspects of the study are novel and important. The study presents a great combination of approaches to understanding anatomical

evolution.

My brief summary of the results: The authors clearly demonstrate the mode of tooth loss in *Pantherophis guttatus* (based on their own sections) and in a few other macrostomatan snakes. They then find indirect evidence (Howship's lacunae on the margins of the pulp cavity) for the same mode of tooth loss in other extant snakes and in fossil *Portugalophis* and *Yurlunggur*, and find histological evidence against the same mode of tooth loss in another extant squamate (*Varanus*). Notably, *Varanus* is dentally similar to snakes in two significant ways: lack of resorption pits and presence of multiple replacement teeth per locus.

The authors make two main claims:

(1) The snake mode of tooth loss evolved by the Jurassic and was present in the common ancestor of crown snakes.

(2) The snake mode of tooth loss is a key innovation in snake evolution.

Additionally, they suggest that (3) the "head-first" model of snake evolution is corroborated by their discoveries.

On point 1:

The conclusion hinges on two elements: (i) the interpretation of Howship's lacunae, and (ii) the assumed phylogenetic position of *Portugalophis* & *Yurlunggur*.

(i) As long as the lacunae are not subject to equifinality (that is, there is no more than one process that can result in the same anatomical feature), then their interpretation is secure. At present, it seems there is a one-to-one relation between lacunae and osteoclast activity in the pulp cavity.

(ii) The importance of the phylogenetic position of *Portugalophis* & *Yurlunggur* arises from the fact that, as Reviewer 2 notes, the authors do not sample a scolecophidian and therefore do not cover the basal node in crown snakes.

The assumption that *Portugalophis* is a basal stem snake has been supported by a number of quasi-independent studies, such as Martill et al. (2015) [based on Zaher & Scanferla 2012], Harrington & Reeder (2017) [based on Hsiang et al. 2015 and Martill et al. 2015] and Garberoglio et al. (2019) [based on Caldwell et al. 2015, which was based on Longrich et al., 2012]. Given published information, there is a reasonable basis for the authors' assumption.

The assumption that *Yurlunggur* is a stem snake is more tenuous, as the authors themselves note. Some studies have found it to be a stem alethinophidian, others a stem snake. The evidence it provides is therefore tenuous.

The issue of *Coniophis* is also importantly raised by Reviewer 2. The first question here is the number of specimens. Regarding maxillary fragments referred to *Coniophis* (cf.) *precedens*, Martill et al. (2015, supplementary information), contra the authors, note some maxillary specimens apparently newly referred to that taxon, including AMNH 53935 [not a reptile in AMNH database; not clear what they intended here], 26660 and 2413 [sic; for 22413 ?]. [The holotype is USNM 2143.] The specimen AMNH FR 22127 (referenced by Reviewer 2) is listed currently as "*Colpodontosaurus* sp.", and based on published illustrations of *Colpodontosaurus*, the Reviewer's re-interpretation of this specimen as *Coniophis* is entirely plausible. That being said, the specimen is not illustrated online, and it doesn't seem to have been published (as the authors note). Thus, I concur with the authors that there is no published evidence for resorption pits in *Coniophis*. Should they later be published, it would confound the authors' conclusions.

The authors correctly point out that, contra Reviewer 2, Zaher & Rieppel (1999) stated that resorption pits do not occur in anomalepidids, but they too err in stating that Zaher & Rieppel (1999) identified nutritive/alveolar foramina in the anomalepidid *Liotyphlops*. In fact, Zaher & Rieppel stated that

"There are no resorption pits and no development of an alveolar foramen" (p. 10) in leptotyphlopids or anomalepidids. That being said, Zaher & Rieppel did identify large nutritive foramina in *Typhlops punctatus*, showing that extant snake dentition is not so homogeneous. Also, I think that the anatomy of the illustrated taxa *Liotyphlops* and *Leptotyphlops* remains incompletely understood. The paper in question aimed to elucidate tooth attachment patterns, not tooth loss.

The authors respond: "However, as discussed above these snakes do not have resorption pits or other signs of external resorption, so their tooth replacement must be the same as in other snakes " Yet, until the present study, external resorption in *Varanus* was unknown, and it remains unknown in *Lanthanotus* and *Heloderma*, which likewise lack resorption pits. The authors' inference that scolecophidians "must" be like alethinophidians is not supported — why could it not be like in *Varanus*, *Lanthanotus* or *Heloderma*? Zaher & Rieppel (1999) did not establish that external resorption sensu the authors does not occur in scolecophidians.

The authors' argument that it's difficult to conduct "destructive" sampling of some scolecophidians (in my view, a poor if widespread description, because histological examination reveals much more than it ruins) is not convincing. If a taxon is necessary, it is necessary. Furthermore, many leptotyphlopids and typhlopids are widespread, listed as "least concern" in the IUCN Red List (e.g., *Rena humilis*), and should be easy enough to come by.

On point 2:

The concept of key innovations is widely recognized but also controversial. The authors do not define exactly what they mean, but typically key innovations are considered in regard to the origin of higher taxa (like snakes), and therefore typically with an adaptive radiation and higher diversification rate. A recent study of diversification rates in squamates generally (Bars-Closel et al., 2017, *Evolution* 71(9): 2243-2261) found no evidence for higher rates of diversification in early-branching (scolecophidian) lineages, challenging the authors' speculation that tooth replacement mode is a key innovation in snakes. If the authors just mean that tooth replacement might be important in the origin of snakes, it might be preferable to dispense with the term "key innovation".

On point 3:

The suggestion of support for the "head-first" model seems overstated. The authors write that their study "reveals that early snake evolution was strongly deterministic around features of the head and dentition." They have (potentially) identified one feature (tooth loss, with speculative adaptive significance), and the postcranium is essentially unknown in Jurassic taxa. The limbs in later Mesozoic snakes are reduced.

Ultimately, given published studies of the phylogenetic position of *Portugalophis*, and the assumption of non-equifinality of Howship's lacunae in the pulp cavity, it is my opinion that the authors have enough data to draw conclusion 1. That being said, the anatomy of scolecophidians (and *Coniophis*) remains a major point of ignorance with the potential to alter that conclusion.

Reviewer #5:

Remarks to the Author:

The manuscript LeBlanc et al. targets several aspects of different modes of tooth replacement in reptiles and uncovered a unique type of replacement in snakes. Authors provide the evidence that ancient snakes possessed an internal form of tooth resorption, similar to their modern relative species. I do not really agree with reviewer #3 that this work is a limited and will target only specialized audience. The manuscript combines histological analyses of several recent reptilian species during different developmental stages as well as fossil snakes' analyses. This study brings a completely new view to comparative odontogenesis and cellular processes contributing to tooth replacement together

with overlap to evolutionary field. Moreover, all statements are well documented by high quality images. Based on these findings, I believe that this manuscript is suitable for NatCom audience.

REVIEWER COMMENTS

Reviewer #1 (Remarks to the Author):

Dear Editor, dear Authors,

I see that the comments I had raised during the first review round have now been addressed.

As a final remark, I would suggest mentioning, somewhere in the last paragraph (lines 411-420) or elsewhere, the recent publication of Zaher et al. (2022) who deals with macrophagy in early snakes, as it appears much relevant to your text about macrophagy.

Zaher, H., D.M. Mohabey, F.G. Grazziotin and J.A.W. Mantilla. 2022. The skull of *Sanajeh indicus*, a Cretaceous snake with an upper temporal bar, and the origin of ophidian wide-gaped feeding. *Zoological Journal of the Linnean Society*

Response: Many thanks for this reference. It is now included in the main text at several points.

In addition, in the sentence about plicidentine in *Varanus* “(the extensive plicidentine at the base of the teeth in *Varanus* partitions the pulp into a series of smaller chambers)” (lines 310-311), I would recommend mentioning also (in the same parenthesis) how is this respective plicidentine morphology in the extinct Eocene palaeowaranid lizard *Palaeowaranus*, which is a distant relative of *Varanus*. See CT-scans of *Palaeowaranus* teeth in Georgalis and Scheyer (2019:fig. 3).

Georgalis, G.L. and T.M. Scheyer. 2019. A new species of *Palaeopython* (Serpentes) and other extinct squamates from the Eocene of Dielsdorf (Zurich, Switzerland). *Swiss Journal of Geosciences* 112:383–417.

Response: We are grateful for the suggestion, but the references we cite here (32, 33) already provide detailed accounts of the occurrence of plicidentine in varanoids.

Also, the last sentence of the text, “Early snakes were likely already macrophagous in the Jurassic, and this dietary specialization may have evolved before the loss of limbs, which typifies the snake bauplan.” seems somehow speculative, particularly when concerning the current non-consensus surrounding the true-snake affinities of the Jurassic forms (particularly *Parviraptor*). Therefore, perhaps some rephrase (or even deleting) of this sentence would be more appropriate.

Response: We have significantly altered the final paragraph and sentence to the manuscript.

Finally, I just point out that there are a few missing spaces appearing in the text (e.g., lines 113, 135) and misspelling of *Malayopython* in line 210 and *Dinilysia* in line 534, and some non-italicization of genera and species names in the references section.

Response: We have addressed these errors.

As such, upon dealing with these ultimate few issues, I would recommend this paper for publication.

Reviewer #2 (Remarks to the Author):

I read carefully the authors rebuttal and thanks their effort in trying to clarify the main issues raised by the referees. Some of my questions were clarified appropriately and I am confident with the answers given. However, I am still not convinced by some of the main arguments put forward in this paper. I am especially puzzled by the resistance from the authors in trying to fill the obvious informational gaps in their work.

Therefore, some of my main criticisms remain unsolved. I will enumerate them below:

1) On scolecophidians – The unique resorption mechanism described by the authors represents the main and really important finding of this study. However, the authors failed to convincingly provide evidence that this mechanism is present in all snakes. Without further evidence from scolecophidians, there is no reason to view the novel mechanism as a synapomorphy of snakes, since it can well be an alethinophidian specialization instead. Filling the informational gap on scolecophidians is therefore crucial to their work. There is actually no published evidence [including Zaher and Rieppel (1999) mentioned by the authors] that addresses specifically the presence/absence of resorption pits in scolecophidians. Therefore, the authors should try to fill that gap of information since it is exactly what their study are meant to clarify. Quoting a sentence from Zaher and Rieppel (1999) simply does not solve the issue (this authors did not provide compelling evidence in that sense). What is seen in *Liotyphlops* can well be a resorption pit instead of an anormally large nutritive foramen.

Response: We have now provided the first detailed account of the tooth replacement cycle in a scolecophidian (pages 12-14 of the revised MS, Fig. 3, Suppl. Fig. 4). Fortunately, we can clarify what the “resorption pits” in the reviewer’s original comments regarding Zaher and Rieppel (1999) are. This has provided important data regarding the unusual and autapomorphic nature of scolecophidian tooth replacement, but it still follows the same sequence of events as other snakes.

Our histological data for a preserved specimen of the typhlopoid *Anilius bicolor* provide evidence for massive amounts of internal resorption in a scolecophidian snake and reveals how the replacement teeth are related to the formation of these abnormally large foramina along the maxilla below the teeth (new Figure 3 and Supplementary information). The “resorption pits” are at first large openings in the maxilla, not the tooth base (and therefore are not resorption pits) and are connected to a massive network of internal resorption within the maxillary bone above the tooth bases. This resorption is modelled in 3-D in Suppl. Fig. 4c.

The replacement teeth form far away from these pits (Fig. 3) along the gumline. Based on the histological sections, modification of the dental pulp and internal resorption begins

prior to the encroachment of the replacement tooth, like in other snakes. This does not occur in varanoids or other squamates, based on our data.

By the time the replacement tooth reaches the base of the functional tooth, the pulp of the latter is completely scalloped internally, similar to other snakes.

Where *Anilius* differs from other snakes is in the encroachment of these pits in the maxilla towards the bases of the teeth, giving the impression of initial external resorption. While we cannot determine why so much resorption occurs within the maxilla and why it expands on to the teeth, the sequence of events in this scolecophidian are the same as other snakes: the internal pulp transforms into a resorptive structure that eats away at the tooth from the inside, before the replacement tooth is ready to replace its predecessor. For example, compare the late resorption stage of *Hydrophis* in Fig. 2d, g, h with the similar stage of the tooth for *Anilius* in Fig. 3j, k.

In other squamates, the insides of these teeth would still be lined with dentine-producing odontoblasts at this stage.

We have also added more alethinophidian specimens to this dataset (main text and supplementary information), increasing our confidence in the prevalence of this internal resorption mechanism in this snake group, as well as confirming that it is the mechanism through which the venom fangs are precisely replaced in viperids (Fig. 3).

The authors further consider that the “extremely small size of scolecophidians” and their rarity in collections prevent the study of tooth morphology on this group. I strongly disagree: scolecophidians are well represented in many major collections, with sometimes huge series of individuals, including *Liotyphlops* species. A significant number of *Typhlops* species grow to large size, and can be accessed for destructive studies if requested appropriately. The fact that the authors insist in avoiding to expand their study to scolecophidian tooth morphology weakens their arguments and conclusions.

Response: We have acknowledged the reviewer’s need to see scolecophidian tooth replacement in more detail and have added this to the impact of this study. We wish to reiterate, however, that sectioning a retracted maxilla and finding teeth at appropriate stages in a scolecophidian is not a trivial task, and we note in the manuscript that this is the way that resorptive structures should be examined in snakes with such extremely small teeth.

2) On the fossils *Portugalophis* and *Yurlunggur* – The authors seem to consider that both fossils are key taxa to their study. However, recent studies have shown that *Yurlunggur* is likely nested within crown Serpentes and does not represent an early diverging snake (e.g., Zaher and Smith, 2020; Zaher et al. 2022). Therefore, it is not surprising to find in *Yurlunggur* a similar internal resorption mechanism than the one described by the authors in crown alethinophidians.

Response: We have acknowledged the contested phylogenetic positioning of *Yurlunggur* in the manuscript already, however, it is nevertheless an important taxon to study and has added important data to this manuscript. To match our text, we have also modified the phylogeny in Fig. 4a to show a polytomy between *Yurlunggur*, *Scolecophidia*, and *Alethinophidia*.

As for *Portugalophis*, I am puzzled by the suggestion from the authors that *Portugalophis* and *Parviraptor* could be in some way distinct lineages, and that any comparison between these two taxa would be misleading. To my knowledge, no one has considered *Portugalophis* and *Parviraptor* in distinct position in the squamate tree either. In that sense, questions raised in the literature regarding the affinities of “parviraptorids” need to be addressed convincingly by the authors. In any case, the presence of internal resorption mechanism is not clearly visible in *Portugalophis* in my opinion. The ct slices and images provided by the authors are not convincing and the ct video does not correspond to the fifth tooth described in the paper but rather to two more posterior dentary teeth. Please, provide compelling evidence through videos and images of the tooth studied and of additional teeth in the dentary.

Response: In the original description of *Portugalophis* (Caldwell et al., 2015), it is recovered in a basal polytomy with other fragmentary Mesozoic snakes. We are unclear of its precise phylogenetic position relative to other “parviraptorids”, however our analysis of its teeth shows that it possesses an internal resorption mechanism that is similar to that of other snakes. This highlights the value of the study and it is agnostic with regards to the other putative snake fossils mentioned by R2, because they have not been tested yet.

At no point in this manuscript were *Parviraptor* or other “parviraptorids” discussed. Criticising the need to address the affinities of “parviraptorids” is simply moving the goalposts of this study beyond what it was designed to do: to determine how snakes replace their teeth and to determine if it is detectable in the fossil record.

For the record: *Yurlunggur* and *Portugalophis* were selected originally because they were isolated jaws (thus able to fit into a micro-CT machine and achieve sufficient resolution) and contained enough preserved teeth to increase the odds of finding ones at the appropriate stages of resorption (e.g., teeth of *Dinilysia* and *Najash* fell away post-mortem, due to their ligamentous tooth attachment). In the process, multiple members of this author list independently found clear scalloping within at least one tooth of the *Portugalophis* dentary and in *Yurlunggur* (see main text and supplementary figures, as well as a new CT slice video we have uploaded for *Portugalophis*, Supplementary Video 8, which corresponds well with the horizontal slice video through an extant *Boa*, Supplementary Video 3). These images and videos provide compelling evidence for internal tooth resorption in our two fossil taxa. Short of thin sections or synchrotron-based nanotomography, we cannot make this any clearer for this reviewer.

No other extant squamate that we are aware of performs this type of tooth replacement, even after we have now examined additional taxa, at this reviewer’s request. This includes examining osteological specimens of *Heloderma* (Suppl. Fig. 11; requested by this reviewer previously) and *Amphisbaena* (Suppl. Fig. 14; based on previously published reports suggesting a lack of resorption pits in amphisbaenians).

In summary, I think the study still suffers from serious flaws and fails to provide all the necessary evidence to support its claims. Additionally, the authors use two questionable fossils in support of their claims (with *Yurlunggur* being demonstrated to be far from the basal divergence of snakes and *Portugalophis* being a poorly preserved specimen with uncertain affinities with snakes). With that in mind, the overall impression is that their taxon sampling seems to be biased to their preferred hypothesis and does not test the broader picture. Finally, I would tend to agree with referee #3 that, even with the inclusion of scolecophidians to their sample, the paper is still written to reach a rather limited and specialized audience.

Response: Regarding the significance of this study: taxon sampling has been expanded and in light of our response to R3's comments, they, and an arbitrating reviewer (R5) have provided supportive comments on the validity and significance of this study.

The original aim of this paper was to determine how snakes replace their teeth and we have accomplished this through detailed histological analysis of the largest comparative sampling of extant snakes to date. We have then tested this in one well-established snake fossil, *Yurlunggur*, and in one of the earliest fossils interpreted as a snake, which is known from partial jaws- the ideal candidate for this study. The resulting scans are consistent with snake-type tooth replacement, and we have interpreted those results accordingly.

There were no biases in the taxon sampling. The sampling was the result of the availability of specimens, CT scanners, histological samples, and COVID restrictions over the last several years. This accusation is unfounded and is not constructive. Scolecophidians are indeed difficult to acquire for destructive sampling from many institutions and as we discovered after several attempts to identify the resorption mechanism in these snakes from CT scanning, histological sampling is in fact necessary with teeth this small. We have now addressed this with valuable new histological data in the revised MS.

Additional References:

Zaher H, Smith KT. 2020. Pythons in the Eocene of Europe reveal a much older divergence of the group in sympatry with boas. *Biology Letters* 16: 20200735.

Zaher H, Mohabey DM, Graziotin FG, Mantilla JAW 2022. The skull of *Sanajeh indicus*, a Cretaceous snake with an upper temporal bar, and the origin of ophidian wide-gaped feeding, *Zoological Journal of the Linnean Society*, 2022;, zlac001, <https://doi.org/10.1093/zoolinlean/zlac001>

Reviewer #3 (Remarks to the Author):

The authors have, in my view, successfully defended the potential interest of their paper to the readership of *Nature Comms* by stressing that the internal pattern of resorption is unique to the clade studied. I find their arguments persuasive. They have also addressed very skilfully the comments of the other two reviewers. I therefore can now recommend publication.

Response: We appreciate the reviewer's re-consideration based on our revised manuscript and responses.

Reviewer #4 (Remarks to the Author):

The manuscript gives a careful study of the manner in which snakes lose their teeth and draws comparisons to lizards. Particularly the histological and histochemical aspects of the study are novel and important. The study presents a great combination of approaches to understanding anatomical evolution.

My brief summary of the results: The authors clearly demonstrate the mode of tooth loss in *Pantherophis guttatus* (based on their own sections) and in a few other macrostomatan snakes. They then find indirect evidence (Howship's lacunae on the margins of the pulp cavity) for the

same mode of tooth loss in other extant snakes and in fossil *Portugalophis* and *Yurlunggur*, and find histological evidence against the same mode of tooth loss in another extant squamate (*Varanus*). Notably, *Varanus* is dentally similar to snakes in two significant ways: lack of resorption pits and presence of multiple replacement teeth per locus.

The authors make two main claims:

(1) The snake mode of tooth loss evolved by the Jurassic and was present in the common ancestor of crown snakes.

(2) The snake mode of tooth loss is a key innovation in snake evolution.

Additionally, they suggest that (3) the “head-first” model of snake evolution is corroborated by their discoveries.

On point 1:

The conclusion hinges on two elements: (i) the interpretation of Howship’s lacunae, and (ii) the assumed phylogenetic position of *Portugalophis* & *Yurlunggur*.

(i) As long as the lacunae are not subject to equifinality (that is, there is no more than one process that can result in the same anatomical feature), then their interpretation is secure. At present, it seems there is a one-to-one relation between lacunae and osteoclast activity in the pulp cavity.

(ii) The importance of the phylogenetic position of *Portugalophis* & *Yurlunggur* arises from the fact that, as Reviewer 2 notes, the authors do not sample a scolecophidian and therefore do not cover the basal node in crown snakes.

Response: We have now had the opportunity to section a scolecophidian snake for this study. This study is now the first to describe the tooth replacement cycle in a scolecophidian. The new data are included in the main text and supplementary information.

The assumption that *Portugalophis* is a basal stem snake has been supported by a number of quasi-independent studies, such as Martill et al. (2015) [based on Zaher & Scanferla 2012], Harrington & Reeder (2017) [based on Hsiang et al. 2015 and Martill et al. 2015] and Garberoglio et al. (2019) [based on Caldwell et al. 2015, which was based on Longrich et al., 2012]. Given published information, there is a reasonable basis for the authors’ assumption.

The assumption that *Yurlunggur* is a stem snake is more tenuous, as the authors themselves note. Some studies have found it to be a stem alethinophidian, others a stem snake. The evidence it provides is therefore tenuous.

Response: We have also clarified this in the revised phylogeny in Fig. 4a.

The issue of *Coniophis* is also importantly raised by Reviewer 2. The first question here is the number of specimens. Regarding maxillary fragments referred to *Coniophis* (cf.) precedens, Martill et al. (2015, supplementary information), contra the authors, note some maxillary specimens apparently newly referred to that taxon, including AMNH 53935 [not a reptile in AMNH database; not clear what they intended here], 26660 and 2413 [sic; for 22413 ?]. [The holotype is USNM 2143.] The specimen AMNH FR 22127 (referenced by Reviewer 2) is listed currently as “*Colpodontosaurus* sp.”, and based on published illustrations of *Colpodontosaurus*, the Reviewer’s re-interpretation of this specimen as *Coniophis* is entirely plausible. That being

said, the specimen is not illustrated online, and it doesn't seem to have been published (as the authors note). Thus, I concur with the authors that there is no published evidence for resorption pits in *Coniophis*. Should they later be published, it would confound the authors' conclusions.

Response: Many thanks for the clarification on this point. However, *Coniophis precedens* Marsh, 1892, is a vertebral form taxon diagnosed around the morphology of a single isolated vertebra (type specimen NMNH/USNM 2143). AMNH 22413 is a separate, isolated, maxillary specimen, and is not a mistake made either by us or by Longrich et al, relative to NMNH/USNM 2143. There is discussion in the published literature regarding the validity of assigning isolated jaw materials to a vertebral taxon created by Marsh in 1892, but suffice it to say that jaws that can unequivocally be assigned to *Coniophis* do not exist, despite Longrich et al.'s referrals. . There is no overlap of anatomy with the type material and two of the maxillae assigned to *Coniophis precedens* by Longrich et al. (2012). These look very different from one another (i.e. are unlikely to represent the same taxon): UCMP 53935 vs AMNH 22413 (Longrich et al. 2012: fig. 2a vs 2k). Because of this taxonomic, and consequently phylogenetic uncertainty, we do not agree that we should be required to sample this taxon for this study.

This does not mean that the isolated jaw elements showing snake features are absent from the Late Cretaceous of North America, nor does it mean that one or more of the jaw fragments figured by Longrich et al. (2012) are not snakes. Rather, what it means is that any concept of *Coniophis* – pending the discovery of a complete specimen showing vertebral features of the type – cannot include dental characters but must be limited to vertebral characters shared with the type specimen.

We acknowledge the possibility that more extensive taxonomic sampling of fossil forms could confound our conclusions on early stem snakes, because transitional morphologies are to be expected in transitional forms; however, this will not change our understanding of the distribution of this feature in extant snakes.

Importantly, if as suggested by our findings in *Portugalophis* and *Yurlunggur* this feature does indeed characterize stem and crown snakes, then it provides an important tool for identifying even fragmentary remains of extinct fossil snakes.

We agree with the reviewer that this can and should be tested in fossils from around the world that have been identified as snakes. The final paragraph of the manuscript now reads:

“Internal resorption can thus join the features characterizing snakes and can be used to identify early members of the group. *Yurlunggur* and other madstoïds have been recovered as a basal snake lineage that predates the divergence of the two major groups of extant snakes (scoleophidia and alethinophidia) by several^{8,9,23} (though not all³⁰) phylogenetic analyses, and *Portugalophis* predates the loss of forelimbs in snakes by about 50 million years^{2,7,49}. Snake-type tooth replacement is therefore a promising feature for identifying putative and even fragmentary snake fossils from the key periods of snake evolutionary history.”

The authors correctly point out that, contra Reviewer 2, Zaher & Rieppel (1999) stated that resorption pits do not occur in anomalepidids, but they too err in stating that Zaher & Rieppel (1999) identified nutritive/alveolar foramina in the anomalepidid *Liotyphlops*. In fact, Zaher & Rieppel stated that “There are no resorption pits and no development of an alveolar foramen” (p. 10) in leptotyphlopids or anomalepidids. That being said, Zaher & Rieppel did identify large

nutritive foramina in *Typhlops punctatus*, showing that extant snake dentition is not so homogeneous. Also, I think that the anatomy of the illustrated taxa *Liotyphlops* and *Leptotyphlops* remains incompletely understood. The paper in question aimed to elucidate tooth attachment patterns, not tooth loss.

The authors respond: “However, as discussed above these snakes do not have resorption pits or other signs of external resorption, so their tooth replacement must be the same as in other snakes” Yet, until the present study, external resorption in *Varanus* was unknown, and it remains unknown in *Lanthanotus* and *Heloderma*, which likewise lack resorption pits. The authors’ inference that scolecophidians “must” be like alethinophidians is not supported — why could it not be like in *Varanus*, *Lanthanotus* or *Heloderma*? Zaher & Rieppel (1999) did not establish that external resorption sensu the authors does not occur in scolecophidians.

Response: We now address this with a detailed examination of tooth resorption in *Anilius bicolor*, a typhlopoid scolecophodian. Their tooth replacement mode is very distinct from varanoids and is the result of extensive (and very deep) internal resorption in the maxilla. While this is unusual by squamate standards, the sequence of resorption in their replacement cycle matches that of other snakes. This is discussed in the revised manuscript in detail.

The authors’ argument that it’s difficult to conduct “destructive” sampling of some scolecophidians (in my view, a poor if widespread description, because histological examination reveals much more than it ruins) is not convincing. If a taxon is necessary, it is necessary. Furthermore, many leptotyphlopids and typhlopids are widespread, listed as “least concern” in the IUCN Red List (e.g., *Rena humilis*), and should be easy enough to come by.

Response: We addressed this with the first ever description of tooth replacement in a scolecophidian. We again note that, as we have learned, the logistical challenge of assessing this is not trivial (e.g., small tooth size, few teeth, awkwardly positioned jaws, precious museum samples), and probably explains why so few histological studies of scolecophidian teeth have been conducted in the past.

On point 2:

The concept of key innovations is widely recognized but also controversial. The authors do not define exactly what they mean, but typically key innovations are considered in regard to the origin of higher taxa (like snakes), and therefore typically with an adaptive radiation and higher diversification rate. A recent study of diversification rates in squamates generally (Bars-Closel et al., 2017, *Evolution* 71(9): 2243-2261) found no evidence for higher rates of diversification in early-branching (scolecophidian) lineages, challenging the authors’ speculation that tooth replacement mode is a key innovation in snakes. If the authors just mean that tooth replacement might be important in the origin of snakes, it might be preferable to dispense with the term “key innovation”.

Response: We have reworded these statements throughout the revised text. In the abstract, this is described as one of the earliest innovations in snakes. Otherwise, the term “key innovation is not used.

On point 3:

The suggestion of support for the “head-first” model seems overstated. The authors write that their study “reveals that early snake evolution was strongly deterministic around features of the head and dentition.” They have (potentially) identified one feature (tooth loss, with speculative adaptive significance), and the postcranium is essentially unknown in Jurassic taxa. The limbs in later Mesozoic snakes are reduced.

Response: These statements have been modified throughout the revised version of the text.

Ultimately, given published studies of the phylogenetic position of *Portugalophis*, and the assumption of non-equifinality of Howship’s lacunae in the pulp cavity, it is my opinion that the authors have enough data to draw conclusion 1. That being said, the anatomy of scolecophidians (and *Coniophis*) remains a major point of ignorance with the potential to alter that conclusion.

Response: We have addressed the concern regarding the nature of scolecophidian tooth replacement. Please see our comment above concerning why we should not be required to sample *Coniophis* for this study, given the sample additions we have made in this latest round of revisions and the original aim of the paper.

Reviewer #5 (Remarks to the Author):

The manuscript LeBlanc et al. targets several aspects of different modes of tooth replacement in reptiles and uncovered a unique type of replacement in snakes. Authors provide the evidence that ancient snakes possessed an internal form of tooth resorption, similar to their modern relative species. I do not really agree with reviewer #3 that this work is a limited and will target only specialized audience. The manuscript combines histological analyses of several recent reptilian species during different developmental stages as well as fossil snakes’ analyses. This study brings a completely a new view to comparative odontogenesis and cellular processes contributing to tooth replacement together with overlap to evolutionary field. Moreover, all statements are well documented by high quality images. Based on these findings, I believe that this manuscript is suitable for NatCom audience.

Response: We thank you for your supportive comments.

Reviewers' Comments:

Reviewer #2:

Remarks to the Author:

I read carefully the revised version provided by the authors and was happy to see that they followed most of my suggestions. The inclusion of a scoleophidian strengthened their argument and filling an important gap in their study. I congratulate the authors for their willingness to expand the study, clarifying most of my main concerns.

Reviewer #4:

Remarks to the Author:

I am satisfied with the revisions in response to my comments. In particular, the inclusion of a "scoleophidian" representative closes a critical gap in knowledge. It turns out the mode of replacement in *Anilius bicolor* is pretty weird, but in my opinion the authors make reasonable arguments that it is essentially similar to examined alethinophidian snakes in that internal resorption is primary and not directly associated with a developing replacement tooth.

I have only a few minor comments.

For *Anilius bicolor*, showing teeth from multiple perspectives (figs. 3d and e) was smart and very useful to the reader.

Line 261 - use an n-dash in "resorption pit-like holes"

The authors identified both resorption pits and separate alveolar foramina in *Anilius bicolor* and on the latter point refer to fig. 3d. Please label the alveolar foramina — it is not clear to me what feature the authors are referring to.

Lines 396-398 - Maybe "must have" is too strong here?

Please look for consistent terminology with regard to the total group of snakes. Line 440 and figure 4 use "Ophidia" but the abstract uses "pan-Serpentes". Normally the crown and "pan" name would be the same, although of course a term like "Ophidia" could apply to a clade less inclusive than Pan-Serpentes but more inclusive than crown Serpentes. Regardless, if Pan-Serpentes is used, cite Head, de Queiroz & Greene, 2020, in *Phylonums* (de Queiroz et al.).

Reviewer #4 (Remarks to the Author):

I am satisfied with the revisions in response to my comments. In particular, the inclusion of a “scolecophidian” representative closes a critical gap in knowledge. It turns out the mode of replacement in *Anilius bicolor* is pretty weird, but in my opinion the authors make reasonable arguments that it is essentially similar to examined alethinophidian snakes in that internal resorption is primary and not directly associated with a developing replacement tooth.

I have only a few minor comments.

For *Anilius bicolor*, showing teeth from multiple perspectives (figs. 3d and e) was smart and very useful to the reader.

Many thanks

Line 261 - use an n-dash in “resorption pit–like holes”

Done

The authors identified both resorption pits and separate alveolar foramina in *Anilius bicolor* and on the latter point refer to fig. 3d. Please label the alveolar foramina — it is not clear to me what feature the authors are referring to.

We have labelled one of the alveolar foramina in the maxilla of *Anilius bicolor* (“af”) in Fig. 3d for clarity.

Lines 396-398 - Maybe “must have” is too strong here?

We have changed this sentence to read “This replacement mode could impart a selective advantage by maintaining a continual supply of functional teeth”.

Please look for consistent terminology with regard to the total group of snakes. Line 440 and figure 4 use “Ophidia” but the abstract uses “pan-Serpentes”. Normally the crown and “pan” name would be the same, although of course a term like “Ophidia” could apply to a clade less inclusive than Pan-Serpentes but more inclusive than crown Serpentes. Regardless, if Pan-Serpentes is used, cite Head, de Queiroz & Greene, 2020, in *Phylogenomics* (de Queiroz et al.).

We have followed the recommendation and clarified the use of Pan-Serpentes (with appropriate citation) throughout the manuscript.